# CONFORMAL PREDICTION FOR DEEP CLASSIFIER VIA TRUNCATING

## ABSTRACT

Conformal Prediction is a distribution-free statistical framework that outputs a set of possible labels to capture the predictive uncertainty. In this work, we show that existing conformal prediction methods may generate inefficient sets arising from the inclusion of redundant labels. To mitigate this issue, we propose a novel conformal prediction algorithm, *Post-Calibration Truncated Conformal Prediction* (PoT-CP), which limits the size of the prediction sets generated by existing conformal prediction methods through a maximum rank cutoff. Specifically, PoT-CP determines this cutoff by minimizing a truncation rank that preserves the marginal coverage of the calibration dataset. The key idea is to eliminate classes with excessive predictive uncertainty, allowing PoT-CP to shorten the prediction sets. Theoretically, we provide the asymptotic validity of marginal coverage for PoT-CP and demonstrate the asymptotic conditional coverage equivalence between PoT-CP and the standard conformal prediction algorithm. Extensive experiments demonstrate that PoT-CP can effectively reduce prediction set sizes while maintaining the stable conditional coverage of various conformal prediction algorithms across different classification tasks.

## 1 INTRODUCTION

Modern neural networks, widely deployed in many high-stakes tasks such as autonomous driving (Bojarski et al., 2016), medical diagnostics (Caruana et al., 2015; Vazquez & Facelli, 2022), and financial decision making (Greenberg & Hershfield, 2019), frequently struggle with unreliable and unexplainable predictions. Therefore, these high-stakes applications require not only precise point predictions but also accurate quantification of predictive uncertainty. In such setting, *Conformal Prediction* (Vovk et al., 2005; Shafer & Vovk, 2008; Balasubramanian et al., 2014; Angelopoulos & Bates, 2021; Manokhin, 2022) provides a promising approach with marginal coverage guarantee to capture predictive uncertainty. Specifically, conformal prediction utilizes a non-conformity score function to produce a finite prediction set, including the ground-truth label with a specified confidence level, for an unseen input data point.

In addition to the marginal coverage, the length of the prediction set refers to the *efficiency*. Higher efficiency (i.e., smaller prediction sets) could speed up the decision-making process or assist users in better assessing the model's reliability (Kagita et al., 2017; Cresswell et al., 2024; De Toni et al., 2024). For instance, in medical diagnosis, smaller prediction sets enable doctors to make faster and more focused decisions while maintaining uncertainty awareness, and in autonomous driving, compact prediction sets allow systems to react more quickly while preserving safety guarantees. Thus, improved efficiency could enhance the practicality of CP in modern machine learning systems, bridging the gap between theoretical guarantees and real-world deployability. Beyond efficiency, *conditional coverage* measures the valid coverage for each data point group, serving as a stronger criterion than marginal coverage. To enhance conditional coverage, Adaptive Prediction Sets (APS) (Romano et al., 2020) compute the non-conformity scores by accumulating sorted softmax probabilities in descending order, which often results in low efficiency. Then, Regularized Adaptive Prediction Sets (RAPS) (Angelopoulos et al., 2020) improves efficiency by penalizing non-conformity scores for unlikely classes, but this improvement is achieved at the expense of conditional coverage, illustrating the inherent trade-off. This motivates our question: *Is it possible to reduce the length of prediction sets and maintain a stable conditional coverage?*

In this work, we empirically show that current non-conformity scores typically lead to inefficient prediction sets, containing redundant classes (see Section 3.1). Specifically, the prediction sets that include the correct labels tend to encompass classes with greater uncertainty compared to the ground-truth labels, thereby undermining efficiency. Furthermore, in prediction sets that exclude the ground-truth label, all included classes are essentially redundant. To address this issue, a potential approach is to remove classes with higher uncertainty than the true label in prediction sets that contain the correct label and to consider prediction sets without the true label as empty. However, determining the appropriate truncation for a test instance poses a challenge, as access to ground truth labels is inherently impossible.

To this end, we propose a novel conformal prediction algorithm, *Post-Calibration Truncated Conformal Prediction* (PoT-CP). Specifically, after employing the existing conformal calibration algorithm, PoT-CP searches for a minimum truncation rank that maintains the marginal coverage of the calibration set. Then, when getting a prediction set from the existing conformal prediction algorithm, PoT-CP truncates the prediction set by the truncation rank. The fundamental principle of PoT-CP is to exclude classes with excessive predictive uncertainty that can be removed without compromising the coverage guarantee. To theoretically understand PoT-CP, we prove that it achieves asymptotic validity in terms of marginal coverage while also asymptotically preserving conditional coverage. Furthermore, we theoretically demonstrate that PoT-CP consistently decreases the length of prediction sets, thereby enhancing their efficiency.

Extensive experiments demonstrate that our method could improve the performance of various existing conformal prediction methods. First, PoT-CP could improve the efficiency of prediction sets generated by different scoring functions within the split conformal prediction framework while preserving conditional coverage. For example, on the ImageNet dataset with APS, PoT-CP reduces the set size of $6.3153$ to $5.4801$, while maintaining a stable conditional coverage. Second, we show that our method not only elevates the performance of different score functions but also enhances other conformal prediction algorithms and general classification tasks.

Our contributions are summarized as follows:

1. We empirically demonstrate that prediction sets from current conformal prediction methods often contain extra classes, resulting in reduced efficiency. Specifically, the over-covered prediction sets contain classes with higher uncertainty compared to the ground-truth label, while under-covered prediction sets should remain empty, devoid of any extraneous label.

2. We introduce a novel conformal prediction algorithm, *Post-Calibration Truncated Conformal Prediction* (PoT-CP). The key idea is to eliminate classes with high uncertainty in the prediction sets. Theoretically, we prove the asymptotic validity of marginal coverage and asymptotic conditional coverage equality.

3. We conduct extensive evaluations to show that PoT-CP improves the performance of existing score functions and various conformal prediction procedures. specifically, we validate that our method can enhance the efficiency of class-wise conformal prediction (Shi et al., 2013) and cluster conformal prediction (Dey et al., 2024).

## 2 PRELIMINARY

In this work, we consider multi-class classification with $K$ classes. Let $\mathcal{X} \subset \mathbb{R}^d$ be the input space and $\mathcal{Y} := \{1, \ldots, K\}$ be the label space. We use $(X, Y) \sim \mathcal{P}_{\mathcal{X}\mathcal{Y}}$ to denote a random data pair satisfying a joint data distribution $\mathcal{P}_{\mathcal{X}\mathcal{Y}}$ and $f : \mathcal{X} \to \mathbb{R}^K$ to denote a classification neural network. Thus, the classifier $\hat{\pi} : \mathcal{X} \to \Delta^{K-1}$ is defined as $\sigma \circ f$, where $\Delta^{K-1}$ is a $(K\text{-}1)$-dimensional probability simplex and $\sigma$ is a normalization function such as the softmax function. Ideally, $\hat{\pi}_y(\boldsymbol{x})$ serves as an approximation to the conditional probability of class $y$ given the image feature $\boldsymbol{x}$, i.e., $\mathbb{P}(Y = y | X = \boldsymbol{x})$. Then, the model prediction is generally made as: $\hat{y} = \arg\max_{y \in \mathcal{Y}} \hat{\pi}_y(\boldsymbol{x})$.

**Conformal Prediction.** Conformal prediction (Balasubramanian et al., 2014; Manokhin, 2022; Angelopoulos et al., 2023) is a statistical framework that generates uncertainty sets containing ground-truth label with a desired probability. In this paper, we mainly focus on *Split Conformal Prediction* which is the most widely-used version of the conformal prediction procedure. Specifically, split conformal prediction divides a given dataset into two disjoint subsets: one for training the base clas-

sifier and the other for conformal calibration. We next outline the main process of split conformal prediction:

1. Divide a given dataset into two disjoint subsets: a training fold $\mathcal{D}_{tr}$ and a calibration fold $\mathcal{D}_{cal}$, with $|\mathcal{D}_{cal}|$ being $n$;

2. Train a deep learning model on the training dataset $\mathcal{D}_{tr}$, and define a non-conformity score function $V(\boldsymbol{x}, y)$;

3. Compute $\widehat{Q}_{1-\alpha}$ as the $\frac{\lceil (n+1)(1-\alpha) \rceil}{n}$ quantile of the calibration scores $\{V(\boldsymbol{x}_i, y_i) : (\boldsymbol{x}_i, y_i) \in \mathcal{D}_{cal}\}$, where $\widehat{Q}_{1-\alpha}$ is defined by

$$\widehat{Q}_{1-\alpha} := \inf\{Q \in \mathbb{R} : \frac{|\{i : V(\boldsymbol{x}_i, y_i) \leq Q\}|}{n} \geq \frac{\lceil (n+1)(1-\alpha) \rceil}{n}\};$$

4. Use the conformal threshold $\widehat{Q}_{1-\alpha}$ to generate a prediction set for a new instance $\boldsymbol{x}_{n+1}$:

$$\mathcal{C}(\boldsymbol{x}_{n+1}) = \{y \in \mathcal{Y} : V(\boldsymbol{x}_{n+1}, y) \leq \widehat{Q}_{1-\alpha}\}. \tag{1}$$

In particular, the score $V(\boldsymbol{x}, y)$ can represent the model's predictive uncertainty for the label $y$. For instance, $V(\boldsymbol{x}, y_2) > V(\boldsymbol{x}, y_1)$ indicates that the model is more confident in predicting $y_1$ than $y_2$ for the instance $\boldsymbol{x}$. Moreover, under the assumption of independent and identically distributed (i.i.d.) data, the prediction set satisfies a formal coverage guarantee for any deep learning model and data distribution. The detailed coverage guarantee is presented below.

**Theorem 1.** *(Conformal coverage guarantee; Papadopoulos et al. (2002)). Suppose $\{(X_i, Y_i)\}_{i=1}^n$ and $(X_{n+1}, Y_{n+1})$ are i.i.d., and define $\widehat{Q}_{1-\alpha}$ as in step 3 above and $\mathcal{C}(X_{n+1})$ as in step 4 above. Then the following holds:*
$$\mathbb{P}(Y_{n+1} \in \mathcal{C}(X_{n+1})) \geq 1 - \alpha.$$

Actually, this inequation is known as *marginal coverage* since it holds in expectation unconditionally across all data points. Furthermore, the validity of marginal coverage shown in Theorem 1 still holds on the assumption of exchangeability (Lei et al., 2018; Tibshirani et al., 2019). The i.i.d. assumption in Theorem 1 is stricter than the assumption of exchangeability that is practicable in the real world.

In the literature on conformal prediction, *conditional coverage* is considered a more strict criterion than the marginal coverage. *Object-conditional coverage*, defined as $\mathbb{P}(Y_{n+1} \in \mathcal{C}(\boldsymbol{x})|X_{n+1}) \geq 1 - \alpha$, is recognized as a common instance of conditional coverage. Although exact object-conditional coverage is theoretically unachievable (Vovk, 2012; Lei & Wasserman, 2014), certain conformal prediction algorithms aim to approximate it. For instance, Adaptive Prediction Sets (APS) (Romano et al., 2020) approximates object-conditional coverage by calculating a non-conformity score based on the cumulative sum of sorted softmax probabilities. Formally, the definition of APS is given by:

$$V_{aps}(\boldsymbol{x}, y, u; \hat{\pi}) := \sum_{y' \in \mathcal{Y}} \hat{\pi}_{y'}(\boldsymbol{x}) \mathbb{1}(r_f(\boldsymbol{x}, y') < r_f(\boldsymbol{x}, y)) + u \cdot \hat{\pi}_y(\boldsymbol{x}), \tag{2}$$

where $r_f(\boldsymbol{x}, y)$ denotes the rank of $\hat{\pi}_y(\boldsymbol{x})$ among the descending softmax probabilities, and $u$ is an independent random variable satisfying a uniform distribution on $[0, 1]$. Although APS demonstrates excellent conditional coverage performance, its efficiency is often significantly lower compared to other methods (Angelopoulos et al., 2020; Ding et al., 2024).

## 3 METHOD

In this section, we give an empirical analysis of the prediction sets from the calibration set. Motivated by this analysis, we explain the proposed method, *Post-Calibration Truncated Conformal Prediction* (PoT-CP), and present its asymptotic analysis along with its provable improvement in predictive efficiency over split conformal prediction.

### 3.1 MOTIVATION

We start with a motivating discussion about the length of prediction sets. As shown in Equation 1, for a given test input $\boldsymbol{x}_{n+1}$, the prediction set includes the labels whose non-conformity score is

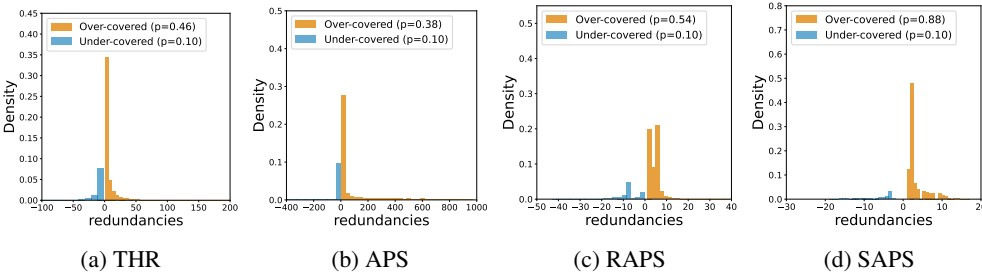

(a) THR        (b) APS        (c) RAPS        (d) SAPS

Figure 1: The redundancy analysis of prediction sets over different scores. "p" represents the proportion of the corresponding part in the total testing dataset. The blue region represents the under-covered prediction sets and the red region represents the over-covered prediction sets.

lower than $\widehat{Q}_{a-\alpha}$. However, some of these labels may have higher predictive uncertainty than the ground truth label, making their inclusion in the prediction set unnecessary. The optimal prediction set should be the smallest set that includes the ground truth label. Similarly, if the prediction set $\mathcal{C}(\boldsymbol{x}_{n+1})$ is under-covered, meaning $y_{n+1} \notin \mathcal{C}(\boldsymbol{x}_{n+1})$, the entire prediction set becomes redundant. In this case, the optimal prediction set would be empty, i.e., $|\mathcal{C}(\boldsymbol{x})| = 0$. This analysis motivates a closer investigation of the redundant labels within prediction sets. To this end, we introduce a new metric to quantify the redundancy of prediction sets. Formally, we define redundancy as follows:

$$\mathrm{Re}(\boldsymbol{x}, y) := \begin{cases} -1 * |\mathcal{C}(\boldsymbol{x})|, & \text{if} \quad |\mathcal{C}(\boldsymbol{x})| - r_V(\boldsymbol{x}, y) < 0, \\ |\mathcal{C}(\boldsymbol{x})| - r_V(\boldsymbol{x}, y), & \text{else}, \end{cases}$$

where $y$ represents the ground truth label of input $\boldsymbol{x}$ and $r_V(\boldsymbol{x}_i, y_i)$ denote the rank of $V(\boldsymbol{x}_i, y_i)$ among of the ascending non-conformity scores $\{V(\boldsymbol{x}_i, y') : y' \in \mathcal{Y}\}$.

Following this, we conduct experiments on split conformal prediction using four score functions based on softmax probabilities: APS, RAPS, Threshold conformal prediction (THR)(Sadinle et al., 2019) and Sorted Adaptive Prediction Sets (SAPS) (Huang et al., 2024). Detailed descriptions of these score functions are provided in Appendix A. Our experiments are conducted on the ImageNet (Deng et al., 2009) dataset, using the pre-trained Inception model from TorchVision (Paszke et al., 2019), with a target error rate of $10\%$ (i.e., $\alpha = 0.1$). For RAPS and SAPS, we tune the hyper-parameters based on the set size. Further details of experiments are provided in Appendix B.

**The redundancies of prediction sets.** In Figure 1, we illustrate the density distribution of redundancies across different score functions. The results show that the majority of prediction sets for various score functions include redundant classes and the proposition of over-covered prediction sets generally is greater than that of under-covered prediction sets. For example, regarding SAPS, the proportion of redundant prediction sets is nearly one. In addition, the length of redundant prediction sets, particularly the under-covered ones, is notably large. For instance, the redundant size in the over-covered prediction sets for APS reaches as high as 750.

From the results, we can design a truncation algorithm that reduces the size of the prediction set while guaranteeing valid marginal coverage. Specifically, for prediction sets that exhibit under-coverage, we can assign them as null sets without compromising the marginal coverage. For over-covered prediction sets, we can truncate the size of the prediction set according to the rank of the ground-truth label. While this idealized truncation approach offers clear benefits, the unknown ground truth labels for test examples make it infeasible. Building on the core principle of rank-based truncation, we propose a post-calibration truncation algorithm that determines appropriate set size cutoffs using information from the split conformal calibration procedure.

### 3.2 POST-CALIBRATION TRUNCATED CONFORMAL PREDICTION

Motivated by the previous analysis, we propose a novel conformal calibration algorithm, *Post-Calibration Truncated Conformal Prediction* (PoT-CP), to find an appropriate maximum set size. The key idea behind our method is to truncate the classes with higher uncertainty than that of the ground-truth label in the prediction sets as much as possible. Specifically, after applying the split conformal prediction algorithm shown in Section 2, we utilize the calibration data to find the max-

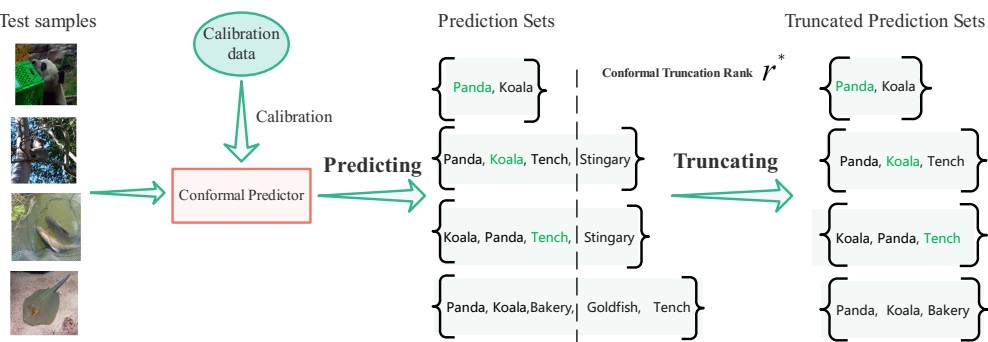

Figure 2: The diagram for Post-Calibration Truncation Conformal Predictor. In the prediction sets, classes with greater uncertainty are assigned a higher rank and the words in green represent the ground-truth label. "$r^*$" represents the truncation rank defined as in Eq. 3.

imum rank of samples whose ground truth label's score is smaller than $\widehat{Q}_{1-\alpha}$. Formally, we define the conformal truncation rank by

$$r^* = \inf\{r \in \mathbb{Z}^+ : \frac{|\{i : V(\boldsymbol{x}_i, y_i) \leq \widehat{Q}_{1-\alpha}\} \cap \{i : r_V(\boldsymbol{x}_i, y_i) \leq r\}|}{n} \geq \frac{\lceil (n+1)(1-\alpha) \rceil}{n}\}, \quad (3)$$

Then, the truncated prediction set can be given by

$$\mathcal{C}_T(\boldsymbol{x}) = \{y \in \mathcal{Y} : V(\boldsymbol{x}, y) \leq \widehat{Q}_{1-\alpha}, r_V(\boldsymbol{x}, y) \leq r^*\}. \quad (4)$$

Particularly, for the non-conformity score functions based on softmax probabilities such as THR and APS, $r_V(\boldsymbol{x}, y)$ is equivalent to $r_f(\boldsymbol{x}, y)$. This implies that a lower conditional probability $\hat{\pi}_y(\boldsymbol{x})$ corresponds to greater uncertainty in assigning $\boldsymbol{x}$ to class $y$.

By using $r^*$ to truncate the prediction sets, we can discard the redundant classes in the prediction sets and maintain the marginal coverage. An illustration of the PoT-CP approach is shown in Figure 2. We now provide some theoretical analysis of PoT-CP under Assumption 1.

**Assumption 1.** *Supposed that $\{(X_1, Y_1), \ldots, (X_n, Y_n), (X_{n+1}, Y_{n+1})\} \in \mathcal{P}_{\mathcal{X}\mathcal{Y}}^{n+1}$ are i.i.d. random variables, where $\{(X_i, Y_i)\}_{i=1}^n$ is a calibration set and $(X_{n+1}, Y_{n+1})$ is a test example.*

**Lemma 1.** *Suppose Assumption 1 holds. $\mathcal{C}(\boldsymbol{x})$ and $\mathcal{C}_T(\boldsymbol{x})$ are defined as in Equation 1 and Equation 4, respectively. Then, we can have that*

$$\mathbb{P}(\lim_{n \to \infty} \mathbb{P}(Y_{n+1} \in \mathcal{C}_T(X_{n+1}) | Y_{n+1} \in \mathcal{C}(X_{n+1})) = 1) = 1.$$

*Moreover, $\mathbb{P}(Y_{n+1} \in \mathcal{C}_T(X_{n+1}) | Y_{n+1} \in \mathcal{C}(X_{n+1}))$ approaches 1 as $n \to \infty$, at a rate of $\sqrt{\frac{\ln n}{n}}$.*

Given the above Lemma 1, we observe that as the calibration size approaches infinity, the probability that the truncated prediction set excludes the ground-truth label converges to zero. Therefore, PoT-CP preserves the marginal coverage. In the following, we further explore the asymptotic equivalence between $\mathcal{C}(X_{n+1})$ and $\mathcal{C}_T(X_{n+1})$.

**Theorem 2.** *(Asymptotic Coverage Equality) Suppose Assumption 1 holds. $\mathcal{C}(\boldsymbol{x})$ and $\mathcal{C}_T(\boldsymbol{x})$ are defined as in Equation 1 and Equation 4. Then, as $n \to \infty$, we have*

$$\mathbb{P}(Y_{n+1} \in \mathcal{C}(X_{n+1}) \iff Y_{n+1} \in \mathcal{C}_T(X_{n+1})) \overset{a.s.}{\to} 1.$$

*This indicates that $Y_{n+1} \in \mathcal{C}(X_{n+1})$ and $Y_{n+1} \in \mathcal{C}_T(X_{n+1})$ are asymptotically equivalent.*

**Corollary 1.** *(Asymptotic Validity of Marginal Coverage) From Theorem 2, we conclude that $\mathbb{P}(\lim_{n \to \infty} \mathbb{P}(Y_{n+1} \in \mathcal{C}_T(X_{n+1}) \geq 1 - \alpha)) = 1$.*

**Corollary 2.** *(Asymptotic conditional-coverage Equality) Suppose Assumption 1 holds. For a test example $(X_{n+1}, Y_{n+1})$, the conditional coverage of $\mathcal{C}(\boldsymbol{x})$ and $\mathcal{C}_T(X_{n+1})$ are asymptotically equivalent. Specifically, we have the following*

1. *Asymptotically Object-conditional coverage: as $n \to \infty$,*

$$\mathbb{P}(\mathbb{P}(Y_{n+1} \in \mathcal{C}(\boldsymbol{x})|X_{n+1} = \boldsymbol{x}) = \mathbb{P}(Y_{n+1} \in \mathcal{C}_T(\boldsymbol{x})|X_{n+1} = \boldsymbol{x})) \overset{a.s.}{\to} 1$$

2. *Asymptotically class-conditional coverage: as $n \to \infty$,*

$$\mathbb{P}(\mathbb{P}(Y_{n+1} \in \mathcal{C}(\boldsymbol{x})|Y_{n+1} = y) = \mathbb{P}(Y_{n+1} \in \mathcal{C}_T(\boldsymbol{x})|Y_{n+1} = y)) \overset{a.s.}{\to} 1$$

Moreover, we can get that the length of the truncated prediction set $\mathcal{C}_T(\boldsymbol{x})$ is not greater than that of the standard prediction sets, i.e., $|\mathcal{C}_T(X_{n+1})| \leq |\mathcal{C}(X_{n+1})|$. The proofs of the theorems and corollaries mentioned above can be found in Appendix C. From Corollary 1 and Corollary 2, we conclude that the truncated prediction set asymptotically equals the prediction set generated by split conformal prediction. We emphasize that PoT-CP is a general framework that can be applied to other conformal prediction algorithms, such as Class-wise Conformal Prediction (Vovk, 2012) and Cluster Conformal Prediction (Ding et al., 2024)(see Section 5).

## 4 EXPERIMENTS

### 4.1 EXPERIMENTAL SETUP

**Datasets and Models.** In main experiments, We consider two common datasets: ImageNet (Deng et al., 2009) and CIFAR-100 (Krizhevsky et al., 2009), both of which are standard benchmarks for conformal prediction. For ImageNet, its test dataset of 50,000 images is divided, allocating 30,000 images to the calibration set and 20,000 images to the test set. The tested models are pre-trained on ImageNet from TorchVision (Paszke et al., 2019). For CIFAR-100, we equally split the test dataset into a calibration set of 5,000 images and a test set of 5,000 images, and the models are fine-tuned based on the pre-trained models from TorchVision.

**Non-conformity score functions.** We assess our method using four non-conformity score functions with $\alpha = 0.1$: THR, APS, RAPS, and SAPS. For score functions that involve hyper-parameters (i.e., RAPS, SAPS), we optimize these parameters through a fine-grained grid search on a subset of the calibration set, referred to as the tuning data, which comprises 20% of the overall calibration data. Additionally, this paper focuses on the split conformal prediction algorithm. Each experiment is repeated 10 times, and we present the average results. Further details regarding the experimental setup are provided in Appendix D. Moreover, Our code is built upon TorchCP (Wei & Huang, 2024).

**Evaluation metrics.** Denote a test dataset by $\{(\boldsymbol{x}_i, y_i)\}_{i=1}^{N_{test}}$. The key metrics for evaluating prediction sets are set size (the average length of prediction sets) and marginal coverage rate (the proportion of test examples whose prediction sets include the ground-truth label). Formally, set size and coverage rate are defined as:

$$\text{Size} = \frac{1}{N_{test}} \sum_{i=1}^{N_{test}} |\mathcal{C}(\boldsymbol{x}_i)|, \qquad \text{Coverage} = \frac{1}{N_{test}} \sum_{i=1}^{N_{test}} \mathbf{1}(y_i \in \mathcal{C}(\boldsymbol{x}_i)).$$

Moreover, We evaluate the conditional coverage of different methods using the following metrics: Worst-slice Coverage (WSC)(Romano et al., 2020), Average Class Coverage Gap (CovGap) (Ding et al., 2024) and Size-stratified Coverage Violation (SSCV) (Angelopoulos et al., 2020). WSC approximates object-conditional coverage, while CovGap measures violations in class-conditional coverage. SSCV captures the adaptiveness of prediction sets in classification tasks. The definition of these metrics can be found in Appendix D.1.

### 4.2 RESULTS

**PoT-CP improves the efficiency of APS while maintaining stable conditional coverage.** In Table 1, we present the results of split conformal prediction and PoT-CP applied to APS. We can observe that the coverage rate of all models is close to the desired coverage $1 - \alpha$. A salient observation is that PoT-CP consistently constructs smaller prediction sets compared to APS. For example, on the Inception model, PoT-CP reduces the size of APS from 35.9576 to 30.6982, with a slight increase in WSC and CovGap. On average, across seven models, the size of APS decreases by approximately

Table 1: Experimental results of ImageNet for APS under $\alpha = 0.1$. "▼" indicates that the average value of our proposed methods is lower than the Base method.

| Models | Coverage | | Size ↓ | | WSC ↓ | | SSCV ↓ | | CovGap ↓ | |
|---|---|---|---|---|---|---|---|---|---|---|
| | APS | +PoT | APS | +PoT | APS | +PoT | APS | +PoT | APS | +PoT |
| ResNeXt101 | 0.899 ±0.0029 | 0.899 ±0.0024 | 7.0472 ±0.2123 | 6.5127 ±0.9696 ▼ | 0.0109 ±0.0088 | 0.0117 ±0.0085 | 0.0678 ±0.0057 | 0.0585 ±0.0191 ▼ | 5.9361 ±0.1387 | 5.9533 ±0.1299 |
| ResNet152 | 0.900 ±0.0038 | 0.900 ±0.0033 | 6.3153 ±0.2212 | 5.4801 ±0.8072 ▼ | 0.0022 ±0.0084 | 0.0026 ±0.0085 | 0.0452 ±0.0065 | 0.0301 ±0.0182 ▼ | 5.4399 ±0.1575 | 5.4740 ±0.1659 |
| ResNet101 | 0.899 ±0.0034 | 0.899 ±0.0030 | 6.8317 ±0.1823 | 6.2196 ±0.8397 ▼ | 0.0095 ±0.0061 | 0.0100 ±0.0061 | 0.0613 ±0.0047 | 0.0490 ±0.0189 ▼ | 5.4116 ±0.1427 | 5.4209 ±0.1364 |
| ResNet50 | 0.900 ±0.0029 | 0.899 ±0.0017 | 9.0544 ±0.2294 | 8.0118 ±1.0763 ▼ | 0.0071 ±0.0089 | 0.0072 ±0.0085 | 0.0626 ±0.0022 | 0.0434 ±0.0201 ▼ | 5.2648 ±0.0863 | 5.2906 ±0.0737 |
| DenseNet161 | 0.899 ±0.0030 | 0.898 ±0.0024 | 6.7598 ±0.2046 | 5.8301 ±0.8827 ▼ | 0.0067 ±0.0057 | 0.0077 ±0.0052 | 0.0582 ±0.0029 | 0.0368 ±0.0212 ▼ | 5.6360 ±0.0919 | 5.6659 ±0.0947 |
| Inception | 0.900 ±0.0028 | 0.900 ±0.0028 | 35.9476 ±1.0070 | 30.6982 ±5.0222 ▼ | 0.0462 ±0.0128 | 0.0513 ±0.0102 | 0.0750 ±0.0020 | 0.0711 ±0.0053 ▼ | 6.5707 ±0.1373 | 6.5821 ±0.1410 |
| ShuffleNet | 0.899 ±0.0028 | 0.899 ±0.0029 | 22.6550 ±0.5211 | 21.0803 ±1.9011 ▼ | 0.0058 ±0.0040 | 0.0067 ±0.0042 | 0.0461 ±0.0030 | 0.0385 ±0.0110 ▼ | 5.6828 ±0.1400 | 5.7037 ±0.1368 |
| **Average** | 0.900 | 0.899 | 13.5159 | **11.9747** | 0.0126 | 0.0139 | 0.0595 | **0.0468** | 5.7060 | 5.7272 |

Table 2: Experimental results of ImageNet for automatic tunning hyper-parameters on SSCV. $\alpha = 0.1$. "▼" indicates that the average value of PoT-CP is lower than the standard method.

| Model | Score | Coverage | | Size | | SSCV | |
|---|---|---|---|---|---|---|---|
| | | Split | +PoT | Split | +PoT | Split | +PoT |
| RAPS | ResNeXt101 | 0.900 ±0.0025 | 0.899 ±0.0028 | 5.2447 ±1.3510 | 4.8561 ±1.4247 ▼ | 0.0614 ±0.0292 | 0.0527 ±0.0262 ▼ |
| | ResNet152 | 0.901 ±0.0025 | 0.900 ±0.0029 | 5.2131 ±0.9996 | 4.6751 ±0.8715 ▼ | 0.0535 ±0.0329 | 0.0360 ±0.0363 ▼ |
| | ResNet101 | 0.900 ±0.0027 | 0.899 ±0.0030 | 4.7839 ±0.9149 | 4.6276 ±1.0253 ▼ | 0.2016 ±0.2690 | 0.1572 ±0.2855 ▼ |
| | ResNet50 | 0.900 ±0.0019 | 0.899 ±0.0022 | 7.0332 ±1.1439 | 6.3667 ±1.0116 ▼ | 0.0427 ±0.0331 | 0.0193 ±0.0147 ▼ |
| | DenseNet161 | 0.900 ±0.0030 | 0.899 ±0.0031 | 5.1367 ±1.2305 | 4.7700 ±1.2232 ▼ | 0.0764 ±0.0249 | 0.0465 ±0.0312 ▼ |
| | Inception | 0.900 ±0.0027 | 0.900 ±0.0027 | 10.5462 ±4.1436 | 10.3672 ±3.9857 ▼ | 0.0625 ±0.0089 | 0.0625 ±0.0089 ▼ |
| | ShuffleNet | 0.900 ±0.0032 | 0.900 ±0.0029 | 12.4808 ±2.6603 | 12.2008 ±2.5904 ▼ | 0.0294 ±0.0180 | 0.0193 ±0.0139 ▼ |
| | **Average** | 0.900 | 0.900 | 7.2055 | **6.8376** | 0.0754 | **0.0562** |
| SAPS | ResNeXt101 | 0.900 ±0.0030 | 0.900 ±0.0031 | 2.6855 ±0.5671 | 2.6684 ±0.5496 ▼ | 0.5551 ±0.4452 | 0.3816 ±0.4462 ▼ |
| | ResNet152 | 0.901 ±0.0022 | 0.900 ±0.0019 | 2.8870 ±0.6570 | 2.8470 ±0.5880 ▼ | 0.0359 ±0.0082 | 0.0371 ±0.0079 ▼ |
| | ResNet101 | 0.900 ±0.0032 | 0.899 ±0.0032 | 2.7634 ±0.0244 | 2.7455 ±0.0383 ▼ | 0.0358 ±0.0041 | 0.0375 ±0.0052 ▼ |
| | ResNet50 | 0.899 ±0.0016 | 0.899 ±0.0017 | 3.4217 ±0.9701 | 3.3957 ±0.9285 ▼ | 0.1363 ±0.2689 | 0.1367 ±0.2688 ▼ |
| | DenseNet161 | 0.900 ±0.0030 | 0.900 ±0.0031 | 3.0163 ±0.6756 | 2.9815 ±0.6536 ▼ | 0.1238 ±0.2728 | 0.0391 ±0.0077 ▼ |
| | Inception | 0.901 ±0.0024 | 0.901 ±0.0024 | 5.6087 ±0.3544 | 5.6069 ±0.3536 ▼ | 0.0093 ±0.0244 | 0.0092 ±0.0244 ▼ |
| | ShuffleNet | 0.900 ±0.0016 | 0.900 ±0.0016 | 5.5509 ±0.2845 | 5.5509 ±0.2846 ▼ | 0.0096 ±0.0261 | 0.0096 ±0.0261 ▼ |
| | **Average** | 0.900 | 0.900 | 3.7048 | **3.6851** | 0.1294 | **0.0930** |

1.55, from 13.52 to 11.97. Additionally, PoT-CP enhances size-conditional coverage (i.e., SSCV). For instance, on the DenseNet161 model, PoT-CP reduces the SSCV of APS from 0.0582 to 0.0368. Overall, PoT-CP improves both the efficiency and SSCV of APS while delivering comparable results in WSC and CovGap relative to standard APS. A similar trend is observed in the CIFAR-100 results, as detailed in Appendix F.

**PoT-CP improves the efficiency of RAPS and SAPS tuned on SSCV.** For the score functions with hyper-parameters, we can optimize them not only based on set size but also by considering SSCV. In this part, we investigate how PoT-CP affects the performance of score functions tuned for SSCV. Specifically, we aim to tune the hyper-parameters of RAPS and SAPS to minimize SSCV within the tuning dataset. Further experimental details are outlined in Appendix D.

Table 2 presents the results of the score function tuned for SSCV on ImageNet, while the results for CIFAR-100 can be found in Appendix F. From Table 2, we can observe that PoT-CP improves the efficiency and reduces the SSCV of prediction sets. For instance, on the ResNet152 model with RAPS, PoT-CP reduces the set size from 5.2131 (Base score) to 4.6751 and lowers SSCV from 0.0535 (Base score) to 0.0360. When averaged across seven models with RAPS, PoT-CP decreases the set size by 0.3679, from a baseline of 7.2055, and reduces SSCV by 0.0192, compared to a baseline of 0.0754. Overall, these results demonstrate that PoT-CP improves the efficiency and reduces the SSCV of split conformal prediction with various score functions, even after prior optimization specifically targeting SSCV.

**Ablation analysis on split conformal prediction.** Here, we provide an empirical analysis of how the scale of the calibration dataset affects the performance of PoT-CP. Specifically, We conduct experiments on the Inception model, varying the calibration set sizes from 1,000 to 20,000 samples, while maintaining the test set fixed at 30,000 samples. We evaluate the performance of split conformal prediction across four different score functions. In addition, we tune the hyper-parameters of RAPS and SAPS based on the average size, with further details provided in Appendix D.

As shown in Figure 3, we can observe that for different score functions, PoT-CP consistently produces smaller prediction sets than the Base score function, regardless of the calibration set size. For

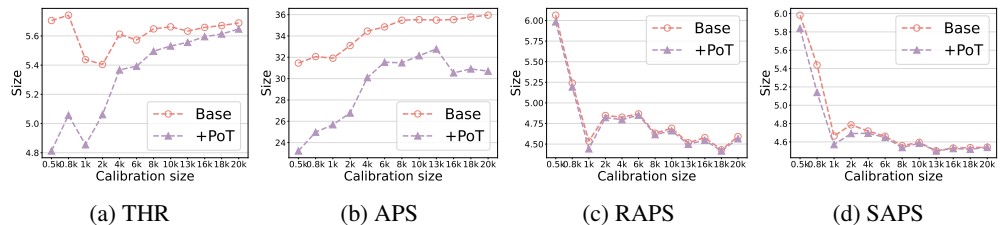

(a) THR       (b) APS       (c) RAPS       (d) SAPS

Figure 3: Effect of the calibration dataset size on average set size for various scores in ImageNet.

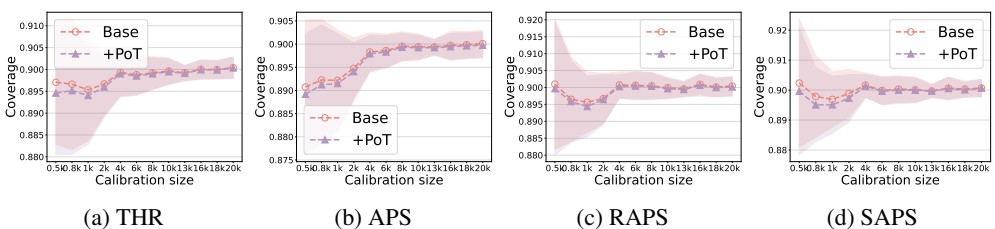

(a) THR       (b) APS       (c) RAPS       (d) SAPS

Figure 4: Effect of the calibration dataset size on empirical coverage for various scores in ImageNet.

Table 3: Experimental results of ImageNet for class-wise conformal prediction under $\alpha = 0.1$. "▼" indicates that the average value of PoT-CP is lower than the baselines.

| Models | Scores | Coverage | | Size | | CovGap | |
|---|---|---|---|---|---|---|---|
| | | Split | +PoT | Split | +PoT | Split | +PoT |
| ResNeXt101 | THR | 0.928 ±0.0021 | 0.917 ±0.0022 | 22.6845 ±3.5244 | 6.7699 ±0.4895 ▼ | 6.6421 ±0.1282 | 6.3175 ±0.1296 |
| | APS | 0.927 ±0.0019 | 0.897 ±0.0017 | 49.6545 ±3.3418 | 6.5153 ±0.4139 ▼ | 6.6358 ±0.1680 | 6.4280 ±0.1384 |
| | RAPS | 0.928 ±0.0014 | 0.903 ±0.0014 | 17.3127 ±1.1241 | 11.4044 ±1.0478 ▼ | 6.5969 ±0.1603 | 6.5333 ±0.1241 |
| | SAPS | 0.928 ±0.0008 | 0.905 ±0.0018 | 17.3114 ±1.1115 | 11.6668 ±0.6817 ▼ | 6.6300 ±0.1546 | 6.6163 ±0.1408 |
| ResNet152 | THR | 0.927 ±0.0024 | 0.917 ±0.0024 | 13.1668 ±1.3695 | 6.4633 ±0.4198 ▼ | 6.6365 ±0.1197 | 6.3878 ±0.1328 |
| | APS | 0.927 ±0.0029 | 0.896 ±0.0027 | 30.6555 ±2.1205 | 5.9883 ±0.4236 ▼ | 6.5965 ±0.0898 | 6.5152 ±0.1245 |
| | RAPS | 0.928 ±0.0023 | 0.900 ±0.0027 | 17.2928 ±1.0959 | 10.7236 ±0.6434 ▼ | 6.5882 ±0.1706 | 6.6157 ±0.1279 |
| | SAPS | 0.928 ±0.0018 | 0.904 ±0.0025 | 17.3817 ±1.1363 | 11.5239 ±0.8821 ▼ | 6.5880 ±0.1329 | 6.6008 ±0.1151 |
| Inception | THR | 0.928 ±0.0019 | 0.911 ±0.0024 | 84.6774 ±4.9129 | 27.8362 ±1.5073 ▼ | 6.7269 ±0.1390 | 6.4653 ±0.1348 |
| | APS | 0.928 ±0.0016 | 0.893 ±0.0021 | 125.0769 ±2.8126 | 23.3743 ±1.1293 ▼ | 6.6847 ±0.1385 | 6.6450 ±0.1127 |
| | RAPS | 0.927 ±0.0019 | 0.903 ±0.0021 | 56.8881 ±1.5086 | 38.4036 ±2.4146 ▼ | 6.6645 ±0.1629 | 6.8307 ±0.1401 |
| | SAPS | 0.928 ±0.0028 | 0.904 ±0.0037 | 56.7532 ±1.4861 | 36.5982 ±2.1672 ▼ | 6.6715 ±0.1588 | 6.8101 ±0.0903 |
| ShuffleNet | THR | 0.928 ±0.0020 | 0.915 ±0.0015 | 39.4263 ±2.1294 | 22.2134 ±1.2105 ▼ | 6.6936 ±0.0898 | 6.4776 ±0.1203 |
| | APS | 0.927 ±0.0014 | 0.890 ±0.0016 | 82.6374 ±2.4700 | 21.0205 ±0.9734 ▼ | 6.7258 ±0.1499 | 6.7877 ±0.1809 |
| | RAPS | 0.928 ±0.0023 | 0.898 ±0.0017 | 51.8426 ±2.1157 | 34.2591 ±2.3800 ▼ | 6.6785 ±0.1575 | 6.8992 ±0.1525 |
| | SAPS | 0.928 ±0.0020 | 0.903 ±0.0021 | 52.0187 ±2.1499 | 33.5909 ±2.0497 ▼ | 6.6520 ±0.1252 | 6.8299 ±0.1990 |
| **Average** | | 0.928 | 0.904 | 45.9238 | **19.2720** | 6.6507 | 6.6100 |

example, with APS, PoT-CP could reduce the set size of APS by nearly 10 across different calibration set sizes. Even with only 1,000 samples in the calibration set, PoT-CP reduces the set size of THR by approximately 0.6. Since RAPS and SAPS are score functions with hyperparameters, these parameters may be less accurately tuned with small calibration sets. Thus, as the calibration set size increases, RAPS and SAPS can achieve smaller prediction set sizes due to better hyperparameter estimation. The results of empirical coverage are presented in Figure 4 and similar results can be observed on the CIFAR-100 dataset, as shown in Appendix F.

## 5 DISCUSSION

**Class-wise conformal prediction.** In the experiments above, we demonstrate that our method could improve the performance of split conformal prediction. Here, we further verify that PoT-CP can enhance the efficiency of prediction sets generated by Class-wise Conformal Prediction (dubbed CCP) (Vovk, 2012; Shi et al., 2013). CCP finds the class-wise quantiles of non-conformity scores on calibration data. Therefore, PoT-CP can be utilized within each class, as explained in Appendix G. We perform experiments with several models on both ImageNet and CIFAR-100.

Table 4: Experimental results of ImageNet for Cluster conformal prediction under $\alpha = 0.1$. "▼" indicates that the average value of PoT-CP is lower than the standard method.

| Models | Scores | Coverage | | Size | | CovGap | |
|---|---|---|---|---|---|---|---|
| | | Split | +PoT | Split | +PoT | Split | +PoT |
| ResNeXt101 | THR | 0.902 ±0.0082 | 0.902 ±0.0081 | 2.1601 ±0.1949 | 2.1226 ±0.1878 ▼ | 5.9354 ±0.1586 | 5.9483 ±0.1565 |
| | APS | 0.901 ±0.0060 | 0.899 ±0.0064 | 8.2605 ±0.6656 | 5.8587 ±0.7552 ▼ | 5.5188 ±0.0976 | 5.5838 ±0.1068 |
| | RAPS | 0.901 ±0.0076 | 0.900 ±0.0077 | 2.8675 ±0.2191 | 2.7803 ±0.2318 ▼ | 5.7404 ±0.1543 | 5.7648 ±0.1473 |
| | SAPS | 0.903 ±0.0067 | 0.902 ±0.0068 | 3.8285 ±1.1118 | 3.8025 ±1.1005 ▼ | 6.8547 ±0.2418 | 6.8662 ±0.2485 |
| ResNet152 | THR | 0.901 ±0.0071 | 0.901 ±0.0071 | 2.1998 ±0.2390 | 2.1804 ±0.2359 ▼ | 6.0166 ±0.1159 | 6.0250 ±0.1129 |
| | APS | 0.903 ±0.0055 | 0.902 ±0.0055 | 7.0071 ±0.4549 | 5.2849 ±0.4722 ▼ | 5.2034 ±0.1549 | 5.2693 ±0.1288 |
| | RAPS | 0.903 ±0.0061 | 0.902 ±0.0060 | 3.6224 ±1.2028 | 3.5379 ±1.2112 ▼ | 5.4265 ±0.1542 | 5.4548 ±0.1543 |
| | SAPS | 0.903 ±0.0071 | 0.903 ±0.0071 | 5.2119 ±1.7445 | 5.1264 ±1.6489 ▼ | 6.5733 ±0.1698 | 6.5782 ±0.1660 |
| Inception | THR | 0.902 ±0.0039 | 0.901 ±0.0041 | 8.8738 ±0.8433 | 7.5645 ±0.8961 ▼ | 5.7870 ±0.1076 | 5.8125 ±0.1091 |
| | APS | 0.903 ±0.0083 | 0.901 ±0.0086 | 41.4338 ±4.4195 | 30.3638 ±4.6957 ▼ | 5.8042 ±0.1866 | 5.8394 ±0.1977 |
| | RAPS | 0.904 ±0.0061 | 0.903 ±0.0061 | 9.3687 ±1.9255 | 8.8053 ±1.5309 ▼ | 6.2317 ±0.1404 | 6.2400 ±0.1452 |
| | SAPS | 0.905 ±0.0048 | 0.904 ±0.0052 | 11.0119 ±3.2620 | 10.0400 ±1.8448 ▼ | 6.7717 ±0.1565 | 6.7798 ±0.1668 |
| ShuffleNet | THR | 0.900 ±0.0071 | 0.899 ±0.0070 | 6.2230 ±0.8786 | 6.1337 ±0.8691 ▼ | 5.8217 ±0.1851 | 5.8402 ±0.1800 |
| | APS | 0.899 ±0.0072 | 0.897 ±0.0077 | 24.0230 ±1.9489 | 20.8449 ±2.5901 ▼ | 5.4355 ±0.1817 | 5.4908 ±0.2027 |
| | RAPS | 0.901 ±0.0076 | 0.900 ±0.0080 | 8.4622 ±1.5824 | 8.2384 ±1.5261 ▼ | 5.8180 ±0.1553 | 5.8444 ±0.1686 |
| | SAPS | 0.902 ±0.0051 | 0.902 ±0.0052 | 9.7420 ±2.7880 | 9.5248 ±2.7841 ▼ | 6.6364 ±0.1970 | 6.6494 ±0.2053 |
| **Average** | | 0.902 | 0.901 | 9.6435 | 8.2631▼ | 5.9735 | 5.9992 |

Table 5: Experimental results for ordinal classification under $\alpha = 0.1$. "▼" indicates that the average value of PoT-CP is lower than the baselines.

| Dataset | Score | Coverage | | Size | | SSCV | |
|---|---|---|---|---|---|---|---|
| | | Base | +PoT | Base | +PoT | Base | +PoT |
| Synthetic | APS | 0.9010±0.0049 | 0.9003±0.0048 | 1.9172±0.0139 | 1.9150±0.0149▼ | 0.1000±0.000 | 0.0865±0.014▼ |
| | THR | 0.9011±0.0048 | 0.9010±0.0048 | 1.7274±0.0143 | 1.7273±0.0142 ▼ | 0.0847±0.019 | 0.0611±0.006▼ |
| UTKFace | APS | 0.9005±0.0066 | 0.8989±0.0059 | 5.5295±0.0773 | 5.4928±0.0769▼ | 0.0443±0.007 | 0.0489±0.009▼ |
| | THR | 0.8996±0.0030 | 0.8991±0.0028 | 5.1416±0.0427 | 5.1326±0.0435▼ | 0.0740±0.001 | 0.0740±0.015▼ |
| **Average** | | 0.9006 | 0.8998 | 3.5789 | **3.5669** | 0.0758 | **0.0676** |

The results for ImageNet are presented in Table 3, while the CIFAR-100 results can be found in Appendix F. we can observe that PoT-CP improves the efficiency of CCP with different score functions while maintaining a stable CovGap. For instance, with the ResNeXt101 model using APS, PoT-CP decreases the set size of CCP from 50.60 (Base score) to 7.28. Additionally, we observe that PoT-CP consistently preserves stable CovGap. Overall, these results demonstrate that PoT-CP effectively reduces the set size of CCP and maintains a stable class-conditional coverage.

**Clutser conformal prediction.** Here, we further verify that PoT-CP still enhances the efficiency of prediction sets generated by Cluster Conformal Prediction (dubbed Cluster CP) (Ding et al., 2024). Cluster CP employs split conformal prediction to clusters of classes in order to achieve cluster-conditional coverage, which serves as an approximation of class-conditional guarantees. Therefore, PoT-CP can be utilized within each cluster, as explained in Appendix H. We perform experiments with several models on both ImageNet and CIFAR-100.

The results for ImageNet are presented in Table 4, while the CIFAR-100 results can be found in Appendix F. A notable finding is that PoT-CP reduces the set size of Cluster CP while maintaining a stable CovGap. For instance, with the ResNeXt101 model using APS, PoT-CP decreases the set size of Cluster CP from 8.2605 (Base score) to 5.8587. Additionally, we observe that PoT-CP consistently preserves stable CovGap. Consequently, these results demonstrate that PoT-CP effectively reduces the set size of Cluster CP without compromising conditional coverage.

**Ordinal classification.** Ordinal classification (Beckham & Pal, 2017), which involves predicting outcomes with a natural order among categories, is a widely used classification task in various applications, such as disease severity labeling (Li et al., 2020) and budget-based recommendations (Septiadi et al., 2018). In this part, we demonstrate the advantages of our method across different ordinal classification tasks. Specifically, we conduct experiments on both a synthetic dataset and a real-world dataset, the UTKFace dataset (Zhang et al., 2017), for age recognition. Further details about the datasets are provided in Appendix I. Then, to train an ordinal classifier, we adopt a standard classifier to generate an unimodal distribution of softmax probabilities across ordinal classes, based on the unimodal framework (Dey et al., 2024).

In Table 5, we present the PoT-CP results across different datasets and various score functions. The results show that PoT-CP can improve the efficiency of prediction sets and significantly reduce SSCV. For example, on the synthetic dataset with THR, PoT-CP reduces the SSCV of 0.1 to 0.0865, representing a 13.5% improvement. Averaged across the two datasets and two score functions, PoT-CP lowers the SSCV from 0.0758 (for the Base score) to 0.0676. Overall, PoT-CP can be applied to a wide range of classification problems and effectively improves SSCV.

## 6 RELATED WORK

Conformal Prediction (CP), a statistical framework characterized by a finite-sample coverage guarantee (Vovk et al., 2005), has recently witnessed a wide adoption in many real-world applications, such as healthcare (Papadopoulos et al., 2009; Hirsch & Goldberger, 2024; Lambert et al., 2024), finance (Wisniewski et al., 2020; Bastos, 2024), robotics (Kuipers et al., 2024; Dixit et al., 2024; Luo et al., 2024) and autonomous systems (Lindemann et al., 2024; Chen et al., 2024).

**Split conformal prediction.** The split conformal prediction framework (Vovk et al., 2005; Shafer & Vovk, 2008; Angelopoulos & Bates, 2021; Manokhin, 2022) is widely applied for classification problems, where the training data set (used to train the base classifier) and calibration data set are disjoint. Various methods aim to design score functions to enhance the efficiency or adaptiveness of prediction sets, including THR (Sadinle et al., 2019), APS (Romano et al., 2020), RAPS (Angelopoulos et al., 2020), SAPS (Huang et al., 2024), and RANK (Luo & Zhou, 2024). However, there is an inherent trade-off between efficiency and conditional coverage. For instance, while APS can closely approximate conditional coverage, it compromises efficiency. Conversely, RAPS improves the efficiency of APS by incorporating a regularization term to reduce the impact of noisy softmax probabilities. Unfortunately, this adjustment in RAPS can degrade the performance in terms of conditional coverage. In this work, we show that PoT-CP can boost the performance of existing score functions in terms of efficiency and maintain a stable conditional coverage.

**Conditional conformal prediction.** Several methods have been proposed to improve the conditional coverage of prediction sets (Vovk, 2012), such as class-conditional coverage (Shi et al., 2013; Sun et al., 2017; Ding et al., 2024; Shi et al., 2024), object-conditional coverage (Sadinle et al., 2019; Melki et al., 2023; Gibbs et al., 2023; Kiyani et al.), and training-conditional coverage (Bian & Barber, 2023; Pournaderi & Xiang, 2024). For instance, to enhance class-conditional coverage, class-conditional conformal prediction (Vovk, 2012) computes quantiles for each class using calibration data. However, when the number of classes is large, limited examples per class can result in inaccurate quantile estimates, producing larger prediction sets. To address this, cluster conformal prediction (Ding et al., 2024) leverages marginal coverage validity within class clusters to approximate class-conditional coverage. In this work, we show that our method enhances the efficiency of various conformal prediction procedures using different non-conformity score functions, all while maintaining stable conditional coverage.

## 7 CONCLUSION AND LIMITATIONS

In this work, we introduce the Post-Calibration Truncated Conformal Predictor (PoT-CP), a truncation method that improves the efficiency of the conformal procedures while preserving the conditional coverage. The key idea of PoT-CP is truncating the classes with high uncertainty in the prediction sets. Extensive experiments show that PoT-CP could improve the efficiency of different score functions. Moreover, PoT-CP is computationally inexpensive and can be easily integrated into existing conformal prediction procedures. Overall, our method is an effective and complementary approach for boosting the efficiency of prediction sets while preserving conditional coverage. We hope that the observations and analyses in this work can inspire more specially designed methods using the truncating technology to improve efficiency.

**Limitations** There remain several limitations in PoT-CP. The i.i.d. assumption is more restrictive than the assumption of exchangeability, which is more appropriate for real-world applications. Furthermore, the theoretical framework is based on asymptotic assumptions, whereas finite-sample theories would likely offer greater practical relevance for the conformal prediction community.

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

## A    DEFINITIONS OF DIFFERENT SCORE FUNCTIONS

**Threshold conformal prediction (THR).** The THR method (Sadinle et al., 2019) measures the similarity between the example and the data space by the conditional probability $\mathbb{P}(Y|X)$. Specifically, given a data pair $(\boldsymbol{x}, y)$, the non-conformity score write as:

$$V_{thr}(\boldsymbol{x}, y; \hat{\pi}) := 1 - \hat{\pi}_y(\boldsymbol{x}). \tag{5}$$

Although yielding small set sizes, THR results in uneven coverage, particularly for difficult classes.

**Regularized Adaptive Prediction Sets (RAPS).** The RAPS method (Angelopoulos et al., 2020) builds on APS by modifying the conformity scores to penalize noisy tail probabilities and regularize the number of samples in the uncertainty set. Specifically, the score function is defined as:

$$V_{raps}(\boldsymbol{x}, y, u; \hat{\pi}) := \sum_{y' \in \mathcal{Y}} \hat{\pi}_{y'}(\boldsymbol{x}) \mathbb{1}(r_f(\boldsymbol{x}, y') < r_f(\boldsymbol{x}, y)) + u \cdot \hat{\pi}_y(\boldsymbol{x}) + \eta \cdot (r_f(\boldsymbol{x}, y) - k_{reg})^+,$$

where $\eta$ represents the weight of regularization, $k_{reg} \geq 0$ are regularization hyper-parameters and $(z)^+$ denotes the positive part of $z$. The regularization term excludes tail probabilities, resulting in smaller prediction sets than APS.

**Sorted Adaptive Prediction Sets (SAPS).** The SAPS method (Huang et al., 2024) mitigates the issue of probability miscalibration by only retaining the highest probability value and discarding all others. Subsequently, the score function is given by:

$$V_{saps}(\boldsymbol{x}, y, u; \hat{\pi}) := \begin{cases} u \cdot \hat{\pi}_{max}(\boldsymbol{x}), & \text{if } r_f(\boldsymbol{x}, y) = 1, \\ \hat{\pi}_{max}(\boldsymbol{x}) + (r_f(\boldsymbol{x}, y) - 2 + u) \cdot \lambda, & \text{else,} \end{cases} \tag{6}$$

where $\lambda$ is a hyper-parameter representing the weight of ranking information and $\hat{\pi}_{max}(\boldsymbol{x})$ denotes the maximum softmax probability. By discarding the unreliable probability values of the models' output, SAPS could further improve the efficiency of prediction sets.

## B    EXPERIMENTAL SETUP OF SECTION 3.1

To analyze the redundancies of prediction sets, we employ four non-conformity scores—THR, APS, RAPS, and SAPS—on the ImageNet dataset. For this analysis, we employ the pre-trained Inception model from TorchVision, setting the target error rate at $10\%$ (i.e., $\alpha = 0.1$). The ImageNet test dataset, consisting of 50,000 images, is split into two subsets: 20,000 images are assigned to the calibration set, while the remaining 30,000 images are used for testing.

**Score functions with hyperparameters.** For the score functions involving hyperparameters, we tune these parameters on a fine grid using a subset of the calibration set designated as the tuning data, which comprises $20\%$ of the calibration data. Let $m$ be the number of data points in the tuning dataset. For RAPS, we follow the experimental setting outlined in previous work (Angelopoulos et al., 2020). We first identify the smallest $k$ such that the top-k predictive sets have coverage at least $\frac{\lceil (m+1)(1-\alpha) \rceil}{m}$ on the tuning dataset, denoting this as $k_{reg}$. We then select $\eta$ from the set $\{0.02, 0.04, \ldots, 0.5\}$ that minimizes the average set size of the $m$ holdout samples. For SAPS, we first apply Temperature Scaling (Guo et al., 2017) on the tuning dataset before choosing $\lambda$ from the same range.

## C    PROOFS

### C.1    PROOF OF LEMMA 1

*Proof.* Given the definition of $\widehat{Q}_{1-\alpha}$, we can have that $Y_{n+1} \in \mathcal{C}(X_{n+1}), V(X_{n+1}, Y_{n+1}) \leq \widehat{Q}_{1-\alpha}$. Then, we reformulate Equation 3 by :

$$\frac{|\{i : V(X_i, Y_i) \leq \widehat{Q}_{1-\alpha}\} \cap \{i : V_f(X_i, Y_i) < r\}|}{n} = \frac{\sum_{i=1}^{n} \mathbb{1}(V(X_i, Y_i) \leq \widehat{Q}_{1-\alpha}) \times \mathbb{1}(V_f(X_i, Y_i) < r)}{n}.$$

With the definition of $\widehat{Q}_{1-\alpha}$, we can have $\frac{\sum_{i=1}^{n} \mathbb{1}(V(X_i,Y_i) \leq \widehat{Q}_{1-\alpha})}{n} \geq \frac{\lceil (n+1)(1-\alpha) \rceil}{n}$ and $\forall Q \leq \widehat{Q}_{1-\alpha}$, $\frac{\sum_{i=1}^{n} \mathbb{1}(V(X_i,Y_i) \leq Q)}{n} < \frac{\lceil (n+1)(1-\alpha) \rceil}{n}$. Thus, for any $X_i \in \{(X_i,Y_i)\}_{i=1}^{n}$, if $V(X_i,Y_i) \leq \widehat{Q}_{1-\alpha}$, $V_f(X_i,Y_i) \leq r^*$. Then, we define $m := n * (1-\alpha)$ as the number of examples whose score is smaller than $\widehat{Q}_{1-\alpha}$.

Define the events: $E_{V_i} : V(X_i,Y_i) \leq \widehat{Q}_{1-\alpha}$ and $E_{W_i} : V_f(X_i,Y_i) \leq r^*$. From the data of $m$ samples, we can evaluate the empirical conditional probability $\widehat{\mathbb{P}}(E_W|E_V)$:

$$\widehat{\mathbb{P}}(E_W|E_V) := \frac{\#\{E_{V_i} \bigcap E_{W_i}\}}{\#E_{V_i}} = 1.$$

By the Law of Large Numbers, as $m \to \infty$, $\widehat{\mathbb{P}}(E_W|E_V) \overset{a.s.}{\to} \mathbb{P}(E_W|E_V)$. Then, we can conclude that $\mathbb{P}(E_W|E_V) = 1$. Finally, for the example $(X_{n+1}, Y_{n+1})$, we can have that as $n \to \infty$, $\mathbb{P}(\{V_f(X_{n+1}, Y_{n+1}) \leq r^*|V(X_{n+1}, Y_{n+1}) \leq \widehat{Q}_{1-\alpha})) \overset{a.s.}{=} 1$.

We provide an analysis of the convergence rate as follows. To quantify the coverage rate, we use Hoeffding's inequality for bounded independent random variables. Let $Z_i$, for $i = 1, 2, \ldots, m$, be independent random variables where $Z_i = 1$ if $E_{W_i}$ occurs given $E_{V_i}$ on the $i$-th instance, and $Z_i = 0$ otherwise. The empirical estimate of the conditional probability is $\widehat{\mathbb{P}}(E_W \mid E_V) = \frac{1}{m} \sum_{i=1}^{m} Z_i$, which takes values in $[0, 1]$. Thus, for any $\epsilon > 0$, we can get that

$$\mathbb{P}(|\mathbb{P}(E_W|E_V) - \widehat{\mathbb{P}}(E_W|E_V)| \geq \epsilon) \leq 2\exp(-2m\epsilon^2).$$

With $\widehat{\mathbb{P}}(E_W|E_V) = 1$, we have

$$\mathbb{P}(\mathbb{P}(E_W|E_V) \leq 1 - \epsilon) \leq 2\exp(-2m\epsilon^2).$$

Let $2\exp(-2m\epsilon^2) = \delta$ and $m \approx (1-\alpha)n$. Then, $\epsilon = \sqrt{\frac{\ln(2/\delta)}{2n(1-\alpha)}}$. Assuming $\delta$ decrease polynomially with $n$, e.g., $\delta = n^{-k}$ where $k > 0$. Thus, the convergence rate becomes:

$$\epsilon = \mathcal{O}(\sqrt{\frac{\ln(n)}{n}}).$$

$\square$

## C.2 PROOF OF THEOREM 2

*Proof.* If $Y_{n+1} \in \mathcal{C}_T(X_{n+1})$, we can have $S(X_{n+1}, Y_{n+1}) \leq \hat{q}$ and $V_f(X_{n+1}, Y_{n+1}) \leq r^*$. Thus, $Y_{n+1} \in \mathcal{C}(X_{n+1})$. Therefore, with Lemma 1, we can have that as $n \to \infty$,

$$\mathbb{P}(Y_{n+1} \in \mathcal{C}(X_{n+1}) \iff Y_{n+1} \in \mathcal{C}_T(X_{n+1})) \overset{a.s.}{\to} 1.$$

. $\square$

## C.3 PROOF OF COROLLARY 1

With the Law of total probability, we can have that

$$\mathbb{P}(Y_{n+1} \in \mathcal{C}_T(X_{n+1})) = \mathbb{P}(Y_{n+1} \in \mathcal{C}_T(X_{n+1})|Y_{n+1} \in \mathcal{C}(X_{n+1}))\mathbb{P}(Y_{n+1} \in \mathcal{C}(X_{n+1}))$$
$$+\mathbb{P}(Y_{n+1} \in \mathcal{C}_T(X_{n+1})|Y_{n+1} \notin \mathcal{C}(X_{n+1}))\mathbb{P}(Y_{n+1} \notin \mathcal{C}(X_{n+1})).$$

By Theorem 1 and Lemma 1, we can have that $\mathbb{P}(\lim_{n\to\infty} \mathbb{P}(Y_{n+1} \in \mathcal{C}_T(X_{n+1}) \geq 1 - \alpha)) = 1$.

## C.4 PROOF OF COROLLARY 2

Firstly, we have that $\mathbb{P}(Y_{n+1} \in \mathcal{C}(\boldsymbol{x}) \mid X_{n+1} = \boldsymbol{x}) = \frac{\mathbb{P}(Y_{n+1} \in \mathcal{C}(\boldsymbol{x}) \cap (X_{n+1} = \boldsymbol{x}))}{\mathbb{P}(X_{n+1} = \boldsymbol{x})}$ and

$\mathbb{P}(Y_{n+1} \in \mathcal{C}_T(\boldsymbol{x}) \mid X_{n+1} = \boldsymbol{x}) = \frac{\mathbb{P}(Y_{n+1} \in \mathcal{C}_T(\boldsymbol{x}) \cap (X_{n+1} = \boldsymbol{x}))}{\mathbb{P}(X_{n+1} = \boldsymbol{x})}$. By Theorem 2, we have that

$$\mathbb{P}(Y_{n+1} \in \mathcal{C}_T(\boldsymbol{x}) \cap (X_{n+1} = \boldsymbol{x})) = \mathbb{P}(Y_{n+1} \in \mathcal{C}(\boldsymbol{x}) \cap (X_{n+1} = \boldsymbol{x})) \quad a.s.$$

Finally, we can conclude that as $n \to \infty$,

$$\mathbb{P}(\mathbb{P}(Y_{n+1} \in \mathcal{C}(\boldsymbol{x})|X_{n+1} = \boldsymbol{x}) = \mathbb{P}(Y_{n+1} \in \mathcal{C}_T(\boldsymbol{x})|X_{n+1} = \boldsymbol{x})) \overset{a.s.}{\to} 1$$

We can use a similar proof to get the asymptotic class-conditional coverage.

### C.5 Proof of Set size from PoT-CP

In this part, we prove that $|\mathcal{C}_T(X_{n+1})| \leq |\mathcal{C}(X_{n+1})|$. Given Equation 4 and Equation 1, for any $y \in \mathcal{C}_T(X_{n+1})$, $V(X_{n+1}, y) \leq \widehat{Q}_{1-\alpha}$ and $V_f(X_{n+1}, y) \leq r^*$. Thus, for any $y \in \mathcal{C}_T(X_{n+1})$, $y \in \mathcal{C}(X_{n+1})$. Thus, we have $\mathcal{C}_T(X_{n+1}) \subseteq \mathcal{C}(X_{n+1})$

## D Experimental setup of Section 4

Here we provide the details of the experimental setup in Section 4.

**Fine-tuning models on the CIFAR-100 dataset**. We fine-tune the pre-trained models from TorchVision (Paszke et al., 2019) using the training dataset of CIFAR-100. The initial learning rate is set to 0.1 and is reduced by a factor of 5 at the 60th, 120th, and 160th epochs. The model is trained for 200 epochs with a batch size of 128, using the SGD optimizer with a weight decay of $5 \times 10^{-4}$ and Nesterov momentum of 0.9.

**Tuning hyper-parameters of score functions.** For score functions that involve hyper-parameters, we tune the hyper-parameters by performing a fine grid search on a subset of the calibration set, referred to as the tuning data, which consists of 20% of the total calibration data. Let $m$ represent the number of data points in the tuning dataset.

1. *Automatically tuning hyper-parameters of RAPS.* We begin by identifying the smallest $k$ such that the top-k predictive sets achieve coverage of at least $\frac{\lceil (m+1)(1-\alpha) \rceil}{m}$ on the tuning dataset, referring to this as $k_{reg}$. To enhance the efficiency of the prediction sets, we then select $\eta$ from the set $\{0.02, 0.04, \ldots, 0.5\}$ that minimizes the average set size of the tuning dataset. In addition, to minimize the SSCV of prediction sets, we choose $\eta$ from the set $\{0.00001, 0.0001, 0.0008, 0.001, 0.0015, 0.002\}$ that minimizes the average SSCV of the $m$ holdout samples.

2. *Automatically tuning hyper-parameters of SAPS.* We first apply Temperature Scaling (Guo et al., 2017) on the tuning dataset. Then, to improve the efficiency of prediction sets, we select $\lambda$ from the set $\{0.02, 0.04, \ldots, 0.5\}$ that minimizes the average set size of the $m$ holdout samples. For SSCV, we search for $\lambda$ within the same range, aiming to minimize the average SSCV of the tuning dataset.

**Class-wise conformal prediction and Cluster conformal prediction.** In the experiments for Class-wise Conformal Prediction and Cluster Conformal Prediction, we use fixed hyper-parameters for RAPS and SAPS. Specifically, for RAPS, we adopt the settings from Cluster Conformal Prediction (Ding et al., 2024), where $k_{reg}$ and $\eta$ are set to 5 and 0.01, respectively. For SAPS, the regularization parameter $\lambda$ is fixed at 0.1.

### D.1 Evaluation metrics for conditional coverage

*Worst-slice Coverage (WSC).* To approximate the Object-conditional coverage in finite samples, the previous works (Romano et al., 2020) measure the worst coverage over different slabs of the feature space. Formally, the definition of WSC is given by:

$$\text{WSC}(\hat{\mathcal{C}}; \delta) = 1 - \alpha - \inf_{v \in \mathbb{R}^d, a < b \in \mathbb{R}} \left\{ \mathbb{P}\left[ Y \in \hat{\mathcal{C}}(X) \mid X \in S_{v,a,b} \right] \text{ s.t. } \mathbb{P}\left[ X \in S_{v,a,b} \right] \geq \delta \right] \right\}$$

where $S_{v,a,b} := \left\{ \boldsymbol{x} \in \mathbb{R}^d : a \leq v^T \boldsymbol{x} \leq b \right\}$ define a slab of the feature space. Therefore, conformal prediction algorithms with better Object-conditional coverage generally achieve lower WSC.

*Average Class Coverage Gap (CovGap).* The CovGap (Ding et al., 2024) quantifies the deviation of the class-conditional coverage from the target coverage level of $1 - \alpha$. let $\mathcal{J}^y := \{i \in [N_{test}]: y_i = y\}$ be the indices of examples with label $y$ and denote the empirical class-conditional coverage of class $y$ by $\hat{c}_y = \frac{\sum_{i \in \mathcal{J}^y} \mathbb{1}\{y_i \in \mathcal{C}(\boldsymbol{x}_i)\}}{|\mathcal{J}^y|}$. We then compute CovGap by:

$$\text{CovGap} = 100 \times \frac{1}{|\mathcal{Y}|} \sum_{y \in \mathcal{Y}} |\hat{c}_y - (1 - \alpha)|.$$

Table 6: Experimental results of ImageNet for APS under $\alpha = 0.1$. "▼" indicates that the average value of our proposed methods is lower than the Base method.

| Models | Coverage | | Size ↓ | | NoE ↓ | | CovS ↓ | | UCovS ↓ | |
|---|---|---|---|---|---|---|---|---|---|---|
| | APS | +PoT | APS | +PoT | APS | +PoT | APS | +PoT | APS | +PoT |
| ResNeXt101 | 0.899 ±0.0029 | 0.899 ±0.0024 | 7.0472 ±0.2123 | 6.5127 ±0.9696 ▼ | 7.2034 ±0.2129 | 6.6578 ±0.9944 ▼ | 7.4099 ±0.2211 | 6.8094 ±1.1014 ▼ | 3.8029 ±0.1461 | 3.8821 ±0.2584 |
| ResNet152 | 0.900 ±0.0038 | 0.900 ±0.0033 | 6.3153 ±0.2212 | 5.4801 ±0.8072 ▼ | 6.4943 ±0.2204 | 5.6360 ±0.8330 ▼ | 6.5253 ±0.2201 | 5.6079 ±0.8999 ▼ | 4.4153 ±0.1930 | 4.3441 ±0.1994 |
| ResNet101 | 0.899 ±0.0034 | 0.899 ±0.0030 | 6.8317 ±0.1823 | 6.2196 ±0.8397 ▼ | 7.0239 ±0.1832 | 6.3947 ±0.8638 ▼ | 7.1135 ±0.1800 | 6.4323 ±0.9510 ▼ | 4.3180 ±0.2273 | 4.3398 ±0.1964 |
| ResNet50 | 0.900 ±0.0029 | 0.899 ±0.0017 | 9.0544 ±0.2294 | 8.0018 ±1.0763 ▼ | 9.3257 ±0.2291 | 8.2425 ±1.1138 ▼ | 9.4699 ±0.2283 | 8.2962 ±1.2102 ▼ | 5.3348 ±0.2027 | 5.4076 ±0.2106 |
| DenseNet161 | 0.899 ±0.0030 | 0.898 ±0.0024 | 6.7598 ±0.2046 | 5.8301 ±0.8827 ▼ | 6.9336 ±0.2058 | 5.9802 ±0.9061 ▼ | 6.9994 ±0.2045 | 5.9754 ±0.9763 ▼ | 4.6332 ±0.2062 | 4.5567 ±0.1253 |
| Inception | 0.900 ±0.0028 | 0.900 ±0.0028 | 35.9476 ±1.1070 | 30.6982 ±5.0222 ▼ | 36.0169 ±0.9985 | 30.7580 ±5.0336 ▼ | 38.7808 ±1.0046 | 32.8949 ±5.6349 ▼ | 10.4091 ±0.6117 | 10.9887 ±0.8485 |
| ShuffleNet | 0.899 ±0.0028 | 0.899 ±0.0029 | 22.6650 ±0.5211 | 21.0803 ±1.9011 ▼ | 23.1075 ±0.5176 | 21.5016 ±1.9396 ▼ | 23.6533 ±0.5273 | 21.8941 ±2.1221 ▼ | 13.7272 ±0.4783 | 13.8393 ±0.4204 |
| **Average** | 0.900 | 0.899 | 13.5159 | **11.9747** | 13.7293 | **12.1673** | 14.2789 | **12.5586** | 6.6629 | 6.7655 |

Table 7: Experimental results of ImageNet for APS under $\alpha = 0.3$. "▼" indicates that the average value of our proposed methods is lower than the Base method.

| Models | Coverage | | Size ↓ | | WSC ↓ | | SSCV ↓ | | CovGap ↓ | |
|---|---|---|---|---|---|---|---|---|---|---|
| | APS | +PoT | APS | +PoT | APS | +PoT | APS | +PoT | APS | +PoT |
| ResNeXt101 | 0.702 ±0.0055 | 0.700 ±0.0050 | 1.5500 ±0.0284 | 1.3786 ±0.1607 ▼ | 0.0058 ±0.0096 | 0.0111 ±0.0101 | 0.2732 ±0.0566 | 0.1870 ±0.1207 ▼ | 8.6969 ±0.1592 | 8.7349 ±0.1795 |
| ResNet152 | 0.702 ±0.0070 | 0.700 ±0.0063 | 1.6897 ±0.0304 | 1.5297 ±0.1346 ▼ | 0.0019 ±0.0090 | 0.0035 ±0.0084 | 0.3000 ±0.0000 | 0.1189 ±0.1251 ▼ | 8.1496 ±0.1015 | 8.1770 ±0.1240 |
| ResNet101 | 0.701 ±0.0059 | 0.699 ±0.0054 | 1.8086 ±0.0279 | 1.6072 ±0.1284 ▼ | 0.0019 ±0.0083 | 0.0041 ±0.0084 | 0.3000 ±0.0000 | 0.0894 ±0.1113 ▼ | 8.1520 ±0.1026 | 8.1994 ±0.1072 |
| ResNet50 | 0.701 ±0.0046 | 0.699 ±0.0047 | 2.2201 ±0.0375 | 1.9598 ±0.2445 ▼ | 0.0023 ±0.0067 | 0.0050 ±0.0056 | 0.1467 ±0.0902 | 0.0807 ±0.0822 ▼ | 8.0130 ±0.0804 | 8.0421 ±0.1027 |
| DenseNet161 | 0.701 ±0.0058 | 0.699 ±0.0055 | 1.7292 ±0.0304 | 1.5704 ±0.1571 ▼ | 0.0046 ±0.0093 | 0.0085 ±0.0085 | 0.3000 ±0.0000 | 0.1700 ±0.1373 ▼ | 8.4238 ±0.1406 | 8.4656 ±0.1495 |
| Inception | 0.702 ±0.0038 | 0.701 ±0.0046 | 2.8826 ±0.1033 | 2.2245 ±0.4631 ▼ | 0.0518 ±0.0097 | 0.0508 ±0.0090 | 0.1852 ±0.0213 | 0.1501 ±0.1048 ▼ | 11.1674 ±0.0690 | 11.1854 ±0.0943 |
| ShuffleNet | 0.700 ±0.0054 | 0.698 ±0.0052 | 3.7849 ±0.0861 | 3.2481 ±0.4296 ▼ | -0.0005 ±0.0058 | 0.0046 ±0.0071 | 0.2308 ±0.0443 | 0.1083 ±0.1142 ▼ | 8.6819 ±0.1199 | 8.7207 ±0.1096 |
| **Average** | 0.701 | 0.699 | 2.2379 | **1.9312** | 0.0097 | 0.0125 | 0.2487 | **0.1292** | 8.7549 | 8.7893 |

Table 8: Experimental results of ImageNet for APS under $\alpha = 0.2$. "▼" indicates that the average value of our proposed methods is lower than the Base method.

| Models | Coverage | | Size ↓ | | WSC ↓ | | SSCV ↓ | | CovGap ↓ | |
|---|---|---|---|---|---|---|---|---|---|---|
| | APS | +PoT | APS | +PoT | APS | +PoT | APS | +PoT | APS | +PoT |
| ResNeXt101 | 0.801 ±0.0039 | 0.798 ±0.0039 | 2.6037 ±0.0505 | 2.0774 ±0.3557 ▼ | 0.0140 ±0.0102 | 0.0157 ±0.0093 | 0.1060 ±0.0263 | 0.0684 ±0.0482 ▼ | 7.7739 ±0.1332 | 7.8463 ±0.1601 |
| ResNet152 | 0.802 ±0.0049 | 0.799 ±0.0048 | 2.7379 ±0.0505 | 2.2738 ±0.2492 ▼ | 0.0008 ±0.0108 | 0.0059 ±0.0090 | 0.0754 ±0.0295 | 0.0474 ±0.0352 ▼ | 7.2394 ±0.0843 | 7.2897 ±0.1045 |
| ResNet101 | 0.800 ±0.0040 | 0.798 ±0.0038 | 2.9778 ±0.0462 | 2.5576 ±0.3664 ▼ | 0.0089 ±0.0068 | 0.0093 ±0.0079 | 0.0609 ±0.0265 | 0.0426 ±0.0282 ▼ | 7.2829 ±0.1549 | 7.3512 ±0.1773 |
| ResNet50 | 0.801 ±0.0041 | 0.799 ±0.0035 | 3.8453 ±0.0770 | 3.2058 ±0.4429 ▼ | 0.0039 ±0.0095 | 0.0054 ±0.0081 | 0.1045 ±0.0181 | 0.0517 ±0.0388 ▼ | 7.0620 ±0.1336 | 7.1223 ±0.1712 |
| DenseNet161 | 0.800 ±0.0052 | 0.798 ±0.0049 | 2.8259 ±0.0611 | 2.4559 ±0.2560 ▼ | 0.0111 ±0.0069 | 0.0132 ±0.0069 | 0.1176 ±0.0247 | 0.0461 ±0.0518 ▼ | 7.5285 ±0.1104 | 7.5637 ±0.1142 |
| Inception | 0.801 ±0.0038 | 0.801 ±0.0037 | 7.4450 ±0.2062 | 6.7802 ±1.1111 ▼ | 0.0638 ±0.0111 | 0.0643 ±0.0106 | 0.1346 ±0.0061 | 0.1268 ±0.0104 ▼ | 9.8631 ±0.1277 | 9.8684 ±0.1267 |
| ShuffleNet | 0.800 ±0.0042 | 0.798 ±0.0041 | 7.8175 ±0.1431 | 6.6036 ±0.8770 ▼ | 0.0041 ±0.0086 | 0.0044 ±0.0094 | 0.0838 ±0.0089 | 0.0452 ±0.0243 ▼ | 7.6809 ±0.0910 | 7.7292 ±0.0932 |
| **Average** | 0.801 | 0.799 | 4.3219 | **3.7078** | 0.0152 | 0.0169 | 0.0975 | **0.0612** | 7.7758 | 7.8244 |

*Size-stratified Coverage Violation (SSCV).* The SSCV metric (Angelopoulos et al., 2020) measures how prediction sets of varying sizes violate conditional coverage. Specifically, given disjoint set-size strata $\{S_j\}_{j=1}^{N_s}$, where $\bigcup_{j=1}^{N_s} S_j = \{1, 2, \cdots, |\mathcal{Y}|\}$, the indices of examples are stratified by prediction set size and denoted as $\mathcal{J}_j = \{i : |\mathcal{C}(\boldsymbol{x}_i)| \in S_j\}$. Formally, *SSCV* is defined by:

$$\text{SSCV} = \sup_j \left| \frac{|\{i \in \mathcal{J}_j : y_i \in \mathcal{C}(\boldsymbol{x}_i)\}|}{|\mathcal{J}_j|} - (1 - \alpha) \right|. \tag{7}$$

# E   ADDITIONAL RESULTS ON IMAGENET

In this section, we provide more analysis about PoT-CP.

**Further analysis of average set size.** To evaluate the prediction set characteristics, we employ four metrics: 'Size' (total average size), 'NoE' (average size excluding empty sets), 'CovS' (average size of sets containing ground-truth labels), and 'UCovS' (average size of sets excluding ground-truth labels). Our experiments on ImageNet using APS with significance level $\alpha = 0.1$ demonstrate that PoT-CP effectively reduces both 'NoE' and 'CovS' measurements while maintaining stable 'UCovS' values, as shown in Table 6.

**Results of APS for different $\alpha$.** To comprehensively evaluate PoT-CP, we conduct experiments using ResNeXt101 as the backbone model on ImageNet across various significance levels $\alpha$. The results are presented in Table 9. We further extend our experiments to multiple models with $\alpha \in \{0.3, 0.2, 0.05, 0.01\}$, as shown in Tables 7, 8, 10, and 11. The results demonstrate that across different significance levels, PoT-CP achieves smaller average set sizes and lower SSCV values while maintaining comparable WSC and CovGap.

Table 9: Experimental results of ImageNet for APS under different $\alpha$ using ResNeXt101 as the backbone model. "▼" indicates that the average value of our proposed methods is lower than the Base method.

| $\alpha$ | Coverage | | Size ↓ | | WSC ↓ | | SSCV ↓ | | CovGap ↓ | |
|---|---|---|---|---|---|---|---|---|---|---|
| | APS | +PoT | APS | +PoT | APS | +PoT | APS | +PoT | APS | +PoT |
| 0.9 | 0.099 ±0.0027 | 0.099 ±0.0027 | 0.1425 ±0.0035 | 0.1425 ±0.0035 ▼ | 0.0011 ±0.0027 | 0.0011 ±0.0027 | 0.0918 ±0.0906 | 0.0843 ±0.0929 ▼ | 4.4873 ±0.1002 | 4.4873 ±0.1002 |
| 0.8 | 0.199 ±0.0049 | 0.199 ±0.0049 | 0.2887 ±0.0062 | 0.2887 ±0.0062 ▼ | 0.0013 ±0.0049 | 0.0013 ±0.0049 | 0.1161 ±0.0670 | 0.1036 ±0.0610 ▼ | 6.0805 ±0.1422 | 6.0805 ±0.1422 |
| 0.7 | 0.298 ±0.0050 | 0.298 ±0.0050 | 0.4410 ±0.0071 | 0.4404 ±0.0070 ▼ | 0.0034 ±0.0057 | 0.0034 ±0.0057 | 0.2390 ±0.0815 | 0.2161 ±0.0936 ▼ | 7.2965 ±0.1837 | 7.2967 ±0.1836 |
| 0.6 | 0.398 ±0.0050 | 0.398 ±0.0050 | 0.6091 ±0.0076 | 0.6088 ±0.0077 ▼ | 0.0046 ±0.0062 | 0.0046 ±0.0062 | 0.1552 ±0.0657 | 0.1552 ±0.0657 ▼ | 8.1102 ±0.2114 | 8.1102 ±0.2114 |
| 0.5 | 0.501 ±0.0047 | 0.500 ±0.0046 | 0.8151 ±0.0099 | 0.7989 ±0.0271 ▼ | 0.0018 ±0.0086 | 0.0020 ±0.0088 | 0.2206 ±0.1944 | 0.1275 ±0.1373 ▼ | 8.6467 ±0.2323 | 8.6523 ±0.2272 |
| 0.4 | 0.602 ±0.0040 | 0.601 ±0.0033 | 1.0913 ±0.0124 | 1.0328 ±0.0653 ▼ | 0.0045 ±0.0108 | 0.0067 ±0.0098 | 0.3346 ±0.1382 | 0.1218 ±0.1480 ▼ | 8.8267 ±0.1399 | 8.8567 ±0.1464 |
| 0.3 | 0.702 ±0.0055 | 0.700 ±0.0050 | 1.5500 ±0.0284 | 1.3786 ±0.1607 ▼ | 0.0058 ±0.0090 | 0.0111 ±0.0101 | 0.2732 ±0.0566 | 0.1870 ±0.1207 ▼ | 8.6969 ±0.1592 | 8.7349 ±0.1795 |
| 0.2 | 0.801 ±0.0039 | 0.798 ±0.0039 | 2.6037 ±0.0505 | 2.0774 ±0.3557 ▼ | 0.0140 ±0.0102 | 0.0157 ±0.0093 | 0.1060 ±0.0263 | 0.0684 ±0.0482 ▼ | 7.7739 ±0.1332 | 7.8463 ±0.1601 |
| 0.1 | 0.899 ±0.0029 | 0.899 ±0.0024 | 7.0472 ±0.2123 | 6.5127 ±0.9696 ▼ | 0.0109 ±0.0088 | 0.0117 ±0.0085 | 0.0678 ±0.0057 | 0.0585 ±0.0191 ▼ | 5.9361 ±0.1387 | 5.9533 ±0.1299 |
| 0.05 | 0.949 ±0.0021 | 0.949 ±0.0018 | 19.8068 ±0.7396 | 17.5855 ±2.7073 ▼ | 0.0083 ±0.0096 | 0.0087 ±0.0096 | 0.0394 ±0.0020 | 0.0345 ±0.0075 ▼ | 4.1618 ±0.1232 | 4.1757 ±0.1103 |
| 0.03 | 0.970 ±0.0009 | 0.970 ±0.0008 | 42.2811 ±1.0037 | 35.1892 ±7.5316 ▼ | 0.0078 ±0.0057 | 0.0085 ±0.0055 | 0.0260 ±0.0008 | 0.0233 ±0.0048 ▼ | 2.9695 ±0.0572 | 2.9883 ±0.0561 |
| 0.01 | 0.990 ±0.0008 | 0.989 ±0.0008 | 145.1734 ±7.0391 | 119.9677 ±24.3730 ▼ | 0.0028 ±0.0084 | 0.0028 ±0.0084 | 0.0099 ±0.0009 | 0.0094 ±0.0013 ▼ | 1.5457 ±0.0512 | 1.5545 ±0.0525 |
| **Average** | - | - | 18.4875 | **15.5019** | 0.0055 | 0.0065 | 0.1400 | **0.0991** | 6.2110 | 6.2281 |

Table 10: Experimental results of ImageNet for APS under $\alpha = 0.05$. "▼" indicates that the average value of our proposed methods is lower than the Base method.

| Models | Coverage | | Size ↓ | | WSC ↓ | | SSCV ↓ | | CovGap ↓ | |
|---|---|---|---|---|---|---|---|---|---|---|
| | APS | +PoT | APS | +PoT | APS | +PoT | APS | +PoT | APS | +PoT |
| ResNeXt101 | 0.949 ±0.0021 | 0.949 ±0.0018 | 19.8068 ±0.7396 | 17.5855 ±2.7073 ▼ | 0.0083 ±0.0096 | 0.0087 ±0.0096 | 0.0394 ±0.0020 | 0.0345 ±0.0075 ▼ | 4.1618 ±0.1232 | 4.1757 ±0.1103 |
| ResNet152 | 0.950 ±0.0019 | 0.949 ±0.0017 | 14.3292 ±0.5110 | 13.5522 ±0.7758 ▼ | 0.0065 ±0.0042 | 0.0057 ±0.0029 | 0.0313 ±0.0019 | 0.0279 ±0.0050 ▼ | 3.9007 ±0.0985 | 3.9025 ±0.0964 |
| ResNet101 | 0.950 ±0.0019 | 0.949 ±0.0017 | 15.7522 ±0.5588 | 14.6741 ±1.3810 ▼ | 0.0049 ±0.0073 | 0.0050 ±0.0072 | 0.0316 ±0.0027 | 0.0275 ±0.0060 ▼ | 3.9371 ±0.1087 | 3.9503 ±0.1008 |
| ResNet50 | 0.950 ±0.0017 | 0.949 ±0.0014 | 20.1592 ±0.5926 | 17.9142 ±2.6869 ▼ | 0.0035 ±0.0091 | 0.0052 ±0.0089 | 0.0327 ±0.0019 | 0.0265 ±0.0080 ▼ | 3.8338 ±0.0904 | 3.8561 ±0.0927 |
| DenseNet161 | 0.950 ±0.0021 | 0.950 ±0.0016 | 16.9994 ±0.7400 | 14.7292 ±2.4749 ▼ | 0.0056 ±0.0050 | 0.0067 ±0.0050 | 0.0331 ±0.0019 | 0.0254 ±0.0110 ▼ | 4.0335 ±0.0812 | 4.0504 ±0.0652 |
| Inception | 0.950 ±0.0015 | 0.950 ±0.0014 | 103.9987 ±1.8489 | 96.1700 ±9.8758 ▼ | 0.0213 ±0.0137 | 0.0213 ±0.0142 | 0.0414 ±0.0033 | 0.0408 ±0.0038 ▼ | 4.1089 ±0.0863 | 4.1167 ±0.0861 |
| ShuffleNet | 0.950 ±0.0023 | 0.949 ±0.0021 | 54.8684 ±1.6728 | 50.1500 ±2.9818 ▼ | 0.0035 ±0.0084 | 0.0041 ±0.0082 | 0.0278 ±0.0014 | 0.0251 ±0.0027 ▼ | 4.0022 ±0.0823 | 4.0114 ±0.0847 |
| **Average** | 0.950 | 0.949 | 35.1305 | **32.1107** | 0.0077 | 0.0081 | 0.0339 | **0.0297** | 3.9969 | 4.0090 |

Table 11: Experimental results of ImageNet for APS under $\alpha = 0.01$. "▼" indicates that the average value of our proposed methods is lower than the Base method.

| Models | Coverage | | Size ↓ | | WSC ↓ | | SSCV ↓ | | CovGap ↓ | |
|---|---|---|---|---|---|---|---|---|---|---|
| | APS | +PoT | APS | +PoT | APS | +PoT | APS | +PoT | APS | +PoT |
| ResNeXt101 | 0.990 ±0.0008 | 0.989 ±0.0008 | 145.1734 ±7.0391 | 119.9677 ±24.3730 ▼ | 0.0028 ±0.0084 | 0.0028 ±0.0084 | 0.0099 ±0.0009 | 0.0094 ±0.0013 ▼ | 1.5457 ±0.0512 | 1.5545 ±0.0525 |
| ResNet152 | 0.990 ±0.0008 | 0.989 ±0.0007 | 76.4870 ±3.6418 | 65.2601 ±8.4096 ▼ | -0.0013 ±0.0069 | -0.0018 ±0.0073 | 0.0085 ±0.0002 | 0.0075 ±0.0011 ▼ | 1.5471 ±0.0519 | 1.5616 ±0.0463 |
| ResNet101 | 0.990 ±0.0007 | 0.989 ±0.0006 | 86.6406 ±3.7504 | 75.2470 ±11.5149 ▼ | 0.0023 ±0.0075 | 0.0019 ±0.0076 | 0.0086 ±0.0006 | 0.0077 ±0.0010 ▼ | 1.5479 ±0.0451 | 1.5619 ±0.0385 |
| ResNet50 | 0.990 ±0.0006 | 0.989 ±0.0006 | 97.9144 ±3.4014 | 83.0018 ±11.9252 ▼ | -0.0005 ±0.0083 | 0.0019 ±0.0083 | 0.0087 ±0.0008 | 0.0082 ±0.0011 ▼ | 1.5466 ±0.0404 | 1.5588 ±0.0405 |
| DenseNet161 | 0.990 ±0.0006 | 0.990 ±0.0006 | 114.0134 ±3.8084 | 93.0432 ±14.2249 ▼ | 0.0094 ±0.0136 | 0.0085 ±0.0142 | 0.0088 ±0.0003 | 0.0078 ±0.0007 ▼ | 1.5416 ±0.0379 | 1.5610 ±0.0364 |
| Inception | 0.989 ±0.0008 | 0.989 ±0.0007 | 368.4748 ±9.6701 | 358.9707 ±12.3052 ▼ | 0.0111 ±0.0081 | 0.0107 ±0.0086 | 0.0194 ±0.0021 | 0.0194 ±0.0021 ▼ | 1.5496 ±0.0498 | 1.5584 ±0.0447 |
| ShuffleNet | 0.990 ±0.0007 | 0.989 ±0.0007 | 229.8350 ±5.9034 | 224.4323 ±6.9871 ▼ | 0.0025 ±0.0073 | 0.0025 ±0.0073 | 0.0114 ±0.0019 | 0.0114 ±0.0019 ▼ | 1.5682 ±0.0490 | 1.5761 ±0.0513 |
| **Average** | 0.990 | 0.989 | 159.7912 | **145.7033** | 0.0038 | 0.0038 | 0.0108 | **0.0102** | 1.5495 | 1.5618 |

# F  RESULTS OF CIFAR-100

In this section, we report the experimental results for CIFAR-100. In particular, Table 12 demonstrates that PoT-CP improves the efficiency and reduces SSCV while maintaining stable conditional coverage for APS applied in split conformal prediction on CIFAR-100. Figure 5 further illustrates that PoT-CP improves efficiency across different score functions and various calibration set sizes. The results of empirical coverage are presented in Figure 6. Additionally, Table 13 provides results for score functions optimized based on SSCV, demonstrating that PoT-CP can reduce both set size and SSCV. Table 14 presents the results of applying PoT-CP to Class-wise CP on the CIFAR-100 dataset. The results show a reduction in set size while maintaining stable class-conditional coverage. Table 15 presents the outcomes of applying PoT-CP to Cluster CP on the CIFAR-100 dataset, showing a consistent decrease in set size while maintaining stable class-conditional coverage.

# G  EMPLOYING POT-CP ON CLASS-WISE CONFORMAL PREDICTION

Let $\mathcal{D}_{cal,j}$ represent the set of calibration examples whose label is $j$. We compute the $1 - \alpha$ qunatile of scores in $\mathcal{D}_{cal,j}$, denoted as $\widehat{Q}_{1-\alpha,j}$. Then, we denote the truncation rank for the class $j$ by $r_j$. The prediction set of class-wise CP, truncated by the truncation ranks, is defined as $\mathcal{C}_r(\boldsymbol{x}) := \{y \in \mathcal{Y} : V(\boldsymbol{x}, y) \leq \widehat{Q}_{1-\alpha,j}, r_V(\boldsymbol{x}, y) \leq r_y\}$. Formally, the conformal truncated rank $r_j^*$ for class $j$ is

Table 12: Experimental results of CIFAR-100 for APS under $\alpha = 0.1$. "▼" indicates that the average value of our proposed methods is lower than the Base method.

| Models | Coverage | | Size ↓ | | WSC ↓ | | SSCV ↓ | | CovGap ↓ | |
|---|---|---|---|---|---|---|---|---|---|---|
| | APS | +PoT | APS | +PoT | APS | +PoT | APS | +PoT | APS | +PoT |
| ResNet101 | 0.898 ±0.0055 | 0.896 ±0.0065 | 4.5710 ±0.1541 | 4.1566 ±0.5925 ▼ | 0.0134 ±0.0243 | 0.0148 ±0.0228 | 0.0575 ±0.0085 | 0.0476 ±0.0141 ▼ | 4.9734 ±0.2502 | 5.0208 ±0.2233 |
| DenseNet161 | 0.898 ±0.0043 | 0.897 ±0.0056 | 7.1957 ±0.2505 | 6.6470 ±0.9057 ▼ | 0.0153 ±0.0131 | 0.0150 ±0.0140 | 0.0767 ±0.0037 | 0.0720 ±0.0098 ▼ | 4.1299 ±0.2922 | 4.1402 ±0.3027 |
| VGG16 | 0.896 ±0.0063 | 0.896 ±0.0062 | 4.6849 ±0.2196 | 4.6288 ±0.1904 ▼ | 0.0259 ±0.0204 | 0.0245 ±0.0227 | 0.0574 ±0.0132 | 0.0587 ±0.0128 ▼ | 5.1213 ±0.3521 | 5.1338 ±0.3507 |
| Inception | 0.899 ±0.0048 | 0.897 ±0.0050 | 10.6457 ±0.2110 | 9.5815 ±1.6948 ▼ | 0.0101 ±0.0186 | 0.0120 ±0.0208 | 0.0777 ±0.0032 | 0.0738 ±0.0074 ▼ | 3.6918 ±0.2070 | 3.7285 ±0.1933 |
| ViT | 0.900 ±0.0084 | 0.897 ±0.0079 | 6.6895 ±0.2485 | 5.9739 ±0.7054 ▼ | 0.0065 ±0.0133 | 0.0117 ±0.0167 | 0.0761 ±0.0102 | 0.0761 ±0.0102 ▼ | 4.2220 ±0.3780 | 4.2435 ±0.3494 |
| **Average** | 0.898 | 0.897 | 6.7574 | **6.1976** | 0.0142 | 0.0156 | 0.0691 | **0.0656** | 4.4277 | 4.4534 |

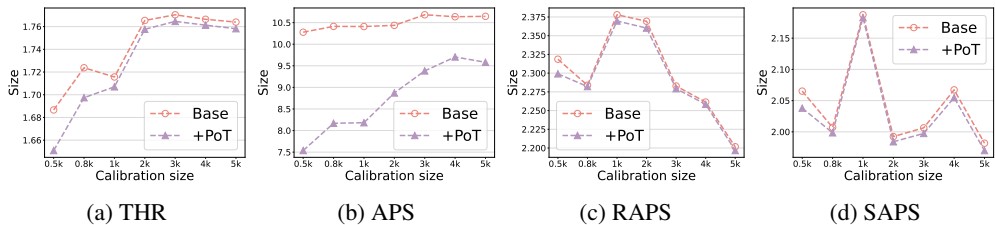

| (a) THR | (b) APS | (c) RAPS | (d) SAPS |
|---|---|---|---|

Figure 5: Effect of the calibration dataset size on average set size for various scores in CIFAR-100.

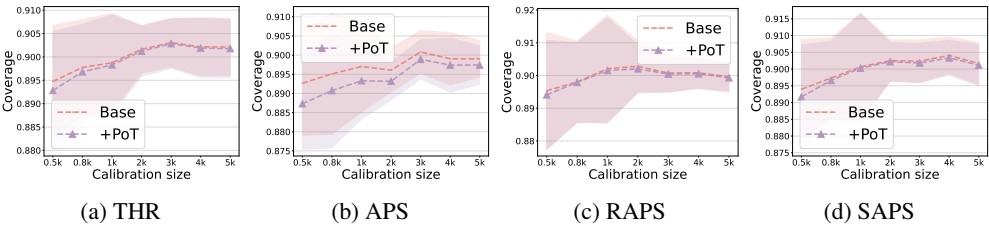

| (a) THR | (b) APS | (c) RAPS | (d) SAPS |
|---|---|---|---|

Figure 6: Effect of the calibration dataset size on empirical coverage for various scores in CIFAR-100.

Table 13: Experimental results of CIFAR-100 for automatic tunning hyper-parameters on SSCV. $\alpha = 0.1$. "▼" indicates that the average value of PoT-CP is lower than the standard method.

| Model | Score | Coverage | | Size | | SSCV | |
|---|---|---|---|---|---|---|---|
| | | Split | +PoT | Split | +PoT | Split | +PoT |
| RAPS | ResNet101 | 0.897 ±0.0062 | 0.897 ±0.0063 | 3.7359 ±0.4387 | 3.5882 ±0.3817 ▼ | 0.0387 ±0.0103 | 0.0360 ±0.0097 ▼ |
| | DenseNet161 | 0.897 ±0.0066 | 0.896 ±0.0076 | 5.5300 ±0.2532 | 5.0613 ±0.6929 ▼ | 0.0607 ±0.0076 | 0.0547 ±0.0118 ▼ |
| | VGG16 | 0.896 ±0.0066 | 0.895 ±0.0061 | 4.5576 ±0.2659 | 4.4726 ±0.2163 ▼ | 0.0622 ±0.0183 | 0.0658 ±0.0163 ▼ |
| | Inception | 0.901 ±0.0050 | 0.899 ±0.0051 | 9.5967 ±0.9102 | 8.9028 ±1.6529 ▼ | 0.0727 ±0.0041 | 0.0704 ±0.0075 ▼ |
| | **Average** | 0.898 | 0.897 | 5.8550 | **5.5062** | 0.0586 | **0.0567** |
| SAPS | ResNet101 | 0.895 ±0.0053 | 0.894 ±0.0057 | 3.4584 ±0.8332 | 3.4005 ±0.8277 ▼ | 0.0568 ±0.0573 | 0.0557 ±0.0585 ▼ |
| | DenseNet161 | 0.897 ±0.0070 | 0.895 ±0.0062 | 2.6484 ±0.2961 | 2.5541 ±0.1685 ▼ | 0.0281 ±0.0115 | 0.0316 ±0.0130 ▼ |
| | VGG16 | 0.896 ±0.0045 | 0.896 ±0.0046 | 4.9702 ±1.0083 | 4.9254 ±0.8985 ▼ | 0.0460 ±0.0384 | 0.0462 ±0.0385 ▼ |
| | Inception | 0.899 ±0.0067 | 0.899 ±0.0064 | 4.1660 ±1.4958 | 3.9185 ±1.2072 ▼ | 0.0374 ±0.0227 | 0.0343 ±0.0197 ▼ |
| | **Average** | 0.897 | 0.896 | 3.8107 | **3.6996** | 0.0421 | **0.0420** |

determined by :

$$r_j^* = \inf\{r \in \mathbb{Z}^+ : \frac{|\{i : V(\boldsymbol{x}_i, j) \leq \widehat{Q}_{1-\alpha}\} \cap \{i : r_V(\boldsymbol{x}_i, j) \leq r\}|}{n} \geq \frac{\lceil(|\mathcal{D}_{cal,j}| + 1)(1 - \alpha)\rceil}{|\mathcal{D}_{cal,j}|}, \boldsymbol{x}_i \in \mathcal{D}_{cal,j}\}.$$

$$(8)$$

Then, the prediction set of a test instance $\boldsymbol{x}$ is then defined by:

$$\mathcal{C}_T(\boldsymbol{x}) := \{y \in \mathcal{Y} : S(\boldsymbol{x}, y) \leq \widehat{Q}_{1-\alpha,j}, V_f(\boldsymbol{x}, y) \leq r_j^*\}.$$

Table 14: Experimental results of CIFAR-100 for class-wise conformal prediction under $\alpha = 0.1$. "▼" indicates that the average value of PoT-CP is lower than the baselines.

| Models | Scores | Coverage | | Size | | CovGap | |
|---|---|---|---|---|---|---|---|
| | | Split | +PoT | Split | +PoT | Split | +PoT |
| ResNet101 | THR | 0.910 ±0.0050 | 0.902 ±0.0053 | 3.2202 ±0.1450 | 2.6506 ±0.0991 ▼ | 4.6956 ±0.3228 | 4.6348 ±0.2785 |
| | APS | 0.908 ±0.0041 | 0.892 ±0.0046 | 6.7448 ±0.1930 | 3.6206 ±0.0749 ▼ | 4.5708 ±0.3496 | 4.7956 ±0.4244 |
| | RAPS | 0.909 ±0.0058 | 0.894 ±0.0053 | 3.7505 ±0.1554 | 3.1386 ±0.1319 ▼ | 4.6473 ±0.2681 | 4.7978 ±0.2891 |
| | SAPS | 0.908 ±0.0057 | 0.896 ±0.0055 | 3.7680 ±0.1724 | 3.3697 ±0.1446 ▼ | 4.6369 ±0.3649 | 4.7969 ±0.3439 |
| DenseNet161 | THR | 0.909 ±0.0086 | 0.903 ±0.0089 | 2.6007 ±0.1900 | 2.2372 ±0.1225 ▼ | 4.8072 ±0.3374 | 4.7718 ±0.2936 |
| | APS | 0.909 ±0.0036 | 0.893 ±0.0041 | 9.2937 ±0.4565 | 4.2271 ±0.2702 ▼ | 4.5657 ±0.2463 | 4.6388 ±0.4015 |
| | RAPS | 0.909 ±0.0042 | 0.894 ±0.0046 | 4.2583 ±0.1082 | 3.0452 ±0.1016 ▼ | 4.6237 ±0.4972 | 4.6485 ±0.5460 |
| | SAPS | 0.910 ±0.0045 | 0.895 ±0.0051 | 3.2620 ±0.1709 | 2.7836 ±0.1428 ▼ | 4.8210 ±0.2752 | 4.8852 ±0.4318 |
| VGG16 | THR | 0.909 ±0.0053 | 0.898 ±0.0050 | 6.1321 ±0.2580 | 5.2184 ±0.2020 ▼ | 4.7071 ±0.4183 | 4.6554 ±0.4436 |
| | APS | 0.909 ±0.0045 | 0.894 ±0.0047 | 7.2640 ±0.3663 | 5.6587 ±0.3385 ▼ | 4.8350 ±0.2892 | 4.8487 ±0.4162 |
| | RAPS | 0.908 ±0.0043 | 0.897 ±0.0044 | 8.6776 ±0.4100 | 7.7704 ±0.4024 ▼ | 4.6199 ±0.3352 | 4.7636 ±0.3356 |
| | SAPS | 0.908 ±0.0039 | 0.898 ±0.0049 | 8.1919 ±0.3269 | 7.3683 ±0.3209 ▼ | 4.6918 ±0.3876 | 4.8120 ±0.3874 |
| Inception | THR | 0.913 ±0.0029 | 0.906 ±0.0033 | 3.5970 ±0.2958 | 2.7412 ±0.1625 ▼ | 4.6130 ±0.3936 | 4.4477 ±0.3920 |
| | APS | 0.910 ±0.0046 | 0.891 ±0.0043 | 12.0023 ±0.1940 | 5.6323 ±0.2041 ▼ | 4.7201 ±0.2779 | 4.8345 ±0.3625 |
| | RAPS | 0.909 ±0.0048 | 0.892 ±0.0040 | 6.3507 ±0.0888 | 4.1556 ±0.1665 ▼ | 4.6464 ±0.2779 | 4.7433 ±0.2869 |
| | SAPS | 0.911 ±0.0025 | 0.898 ±0.0025 | 3.7521 ±0.2270 | 3.3180 ±0.1968 ▼ | 4.5221 ±0.3809 | 4.6212 ±0.3490 |
| | **Average** | 0.909 | 0.896 | 5.8041 | **4.1835** | 4.6702 | 4.7310 |

Table 15: Experimental results of CIFAR-100 for Cluster conformal prediction under $\alpha = 0.1$. "▼" indicates that the average value of PoT-CP is lower than the standard method.

| Models | Scores | Coverage | | Size | | CovGap | |
|---|---|---|---|---|---|---|---|
| | | Split | +PoT | Split | +PoT | Split | +PoT |
| ResNet101 | THR | 0.898 ±0.0091 | 0.898 ±0.0092 | 2.3118 ±0.1862 | 2.2894 ±0.1873 ▼ | 4.6943 ±0.5328 | 4.6933 ±0.5229 |
| | APS | 0.903 ±0.0046 | 0.901 ±0.0045 | 5.1971 ±0.3104 | 4.6409 ±0.3166 ▼ | 4.7425 ±0.2137 | 4.7564 ±0.2175 |
| | RAPS | 0.902 ±0.0039 | 0.901 ±0.0040 | 2.8672 ±0.0595 | 2.8367 ±0.0597 ▼ | 4.7083 ±0.2526 | 4.7424 ±0.2392 |
| | SAPS | 0.899 ±0.0063 | 0.899 ±0.0064 | 2.8869 ±0.2814 | 2.8856 ±0.2817 ▼ | 5.0372 ±0.3527 | 5.0394 ±0.3453 |
| DenseNet161 | THR | 0.902 ±0.0099 | 0.902 ±0.0100 | 1.9952 ±0.1267 | 1.9843 ±0.1243 ▼ | 4.4526 ±0.3867 | 4.4451 ±0.3819 |
| | APS | 0.901 ±0.0068 | 0.900 ±0.0071 | 7.6815 ±0.3913 | 6.7549 ±0.3532 ▼ | 4.1766 ±0.3401 | 4.1972 ±0.3281 |
| | RAPS | 0.901 ±0.0087 | 0.900 ±0.0092 | 3.8745 ±0.1408 | 3.7908 ±0.1549 ▼ | 4.1283 ±0.4117 | 4.1453 ±0.4180 |
| | SAPS | 0.903 ±0.0060 | 0.903 ±0.0059 | 2.4715 ±0.4742 | 2.4698 ±0.4748 ▼ | 4.7311 ±0.3615 | 4.7334 ±0.3571 |
| VGG16 | THR | 0.899 ±0.0111 | 0.898 ±0.0107 | 4.2667 ±0.4498 | 4.1878 ±0.4356 ▼ | 4.4859 ±0.4469 | 4.4889 ±0.4737 |
| | APS | 0.902 ±0.0086 | 0.901 ±0.0087 | 5.4036 ±0.5525 | 5.2537 ±0.5543 ▼ | 4.6645 ±0.3385 | 4.6887 ±0.3244 |
| | RAPS | 0.901 ±0.0077 | 0.901 ±0.0077 | 5.6311 ±0.7096 | 5.6007 ±0.7014 ▼ | 4.7868 ±0.3904 | 4.8161 ±0.3809 |
| | SAPS | 0.901 ±0.0076 | 0.901 ±0.0076 | 6.1813 ±0.7256 | 6.1590 ±0.7213 ▼ | 4.8167 ±0.3597 | 4.8072 ±0.3416 |
| Inception | THR | 0.901 ±0.0066 | 0.901 ±0.0065 | 2.1901 ±0.1463 | 2.1616 ±0.1389 ▼ | 4.2990 ±0.4605 | 4.2830 ±0.4575 |
| | APS | 0.903 ±0.0063 | 0.901 ±0.0062 | 10.9725 ±0.3840 | 9.7192 ±0.5173 ▼ | 3.9609 ±0.1483 | 3.9745 ±0.1223 |
| | RAPS | 0.900 ±0.0078 | 0.899 ±0.0077 | 5.7199 ±0.1694 | 5.5125 ±0.1544 ▼ | 3.9817 ±0.1640 | 3.9906 ±0.1597 |
| | SAPS | 0.905 ±0.0070 | 0.905 ±0.0071 | 2.7252 ±0.3980 | 2.7233 ±0.3979 ▼ | 4.8873 ±0.2414 | 4.8944 ±0.2481 |
| | **Average** | 0.901 | 0.901 | 4.5235 | 4.3106 ▼ | 4.5346 | 4.5435 |

## H  EMPLOYING PoT-CP ON CLUSTER CONFORMAL PREDICTION

We begin by using the tuning dataset to train a clustering function $h$ which maps each class $y \in \mathcal{Y}$ to one of $m$ clusters. For the selection of $m$, we follow the clustering algorithm from cluster CP (Ding et al., 2024).

Let $\mathcal{D}_{cal,j}$ represent the set of calibration examples whose labels belong to the cluster $j$. We compute the $1 - \alpha$ quantile of scores in $\mathcal{D}_{cal,j}$, denoted as $\widehat{Q}_{1-\alpha,j}$. Then, the calibration set $\mathcal{D}_{cal,j}$ is treated as a dataset for a multi-class problem, where the classes are restricted to those within cluster $j$. Similar to Section 3.2, we employ PoT-CP to each cluster to obtain a truncated prediction set. Then, the truncated prediction set for the cluster $j$ is denoted as $\mathcal{C}_{T,j}(\boldsymbol{x})$. Finally, the prediction set of a test instance $\boldsymbol{x}$ is then defined by:

$$\mathcal{C}_T(\boldsymbol{x}) := \bigcup_{j=1}^{m} \mathcal{C}_{T,j}(\boldsymbol{x}).$$

# I  ORDINAL CLASSIFICATION

In this section, we conduct experiments using a synthetic dataset and the UTKFace dataset, adhering to the experimental setup of the prior research (Xu et al., 2023). The details are provided below.

**Synthetic Dataset.** We generated a 10-class ordinal dataset on a 2-D plane, with each class containing 2,000 data points drawn from a Gaussian distribution. The $i$-th class is centered at the coordination $[i, i]$, with a randomly generated covariance matrix. Then, The dataset was divided into 4,000 points for the training set, 8,000 for the calibration set, and 8,000 for the test set. For classification, we implemented a two-layer MLP with 50 neurons in the hidden layer.

**UTKFace.** The UTKFace dataset comprises over 20,000 images, spanning ages from 0 to 116 years. Each image is annotated with the individual's age. In our analysis, we include only individuals under 100 years old and discretize age into 20 groups, each covering a 5-year range (e.g., group 0 for ages 0-4, group 1 for ages 5-9, etc.). Moreover, we train an ordinal classifier based on the ResNet34 (He et al., 2016).

