# OpenReview forum: "Conformal Prediction for Deep Classifier via Truncating"
_ICLR.cc/2025/Conference — Submitted to ICLR 2025_

### Official Review · Reviewer_s7iL · 2024-10-26

**Soundness:** 2
**Presentation:** 3
**Contribution:** 2
**Rating:** 5
**Confidence:** 4

**Summary:**

This paper aims to improve the efficiency of conformal prediction sets by "truncating redundant labels," where the authors refer to this redundancy as predicted labels contained in the prediction set although these do not include the ground truth. The key idea is to find a ranking parameter r* using the calibration data that truncates the set sizes while maintaining coverage. The authors proved an asymptotic marginal coverage and asymptomatic conditional coverage (if the base conformal method achieves it). Numerical experiments show that the proposed method reduces the set sizes, maintaining adaptivity, and coverage.

**Strengths:**

1. A plug-in method that can be used in combination with any conformal approach to reduce the set size.
2. The authors proved a form of `no harm': when the sample size n goes to infinity the truncated set and the original conformal set would contain the test label.
3. To the best of my understanding, there are no additional hyper-parameters required to be tuned to apply the truncation; there is a procedure to tune r*.
4. Numerical experiments demonstrate the gain in performance in the sense that the constructed sets are smaller in size, adaptive, and achieve the desired coverage rate.

**Weaknesses:**

1. Marginal coverage is only guaranteed asymptotically, which arguably undermines the appeal of conformal prediction. See question (1) below.
2. When the sample size n goes to infinity, I expect the prediction sets constructed by the proposed method to get closer and closer to the sets constructed by the base conformal method. For an oracle model for which the conditional class probabilities are known, following Eq. (3), the rank r* is anticipated to get higher as n increases up to a point where the base and truncated methods would coincide. This behavior is somewhat demonstrated in Figure 3. As a result, it is not entirely clear to me why would one be interested in implementing this method. In the regime where it is provably valid, it almost coincides with the original score function.

**Questions:**

1. To address the problem of asymptotic marginal coverage, would it be possible to apply an additional calibration step to tune r*? That is, the first calibration step amounts to finding Q and r* (as happens currently), and the second calibration step aims to find a Q* (based on a fresh calibration set) as in standard conformal.

2. Line 765: what do you mean by P[Ev|Ev] ? I believe this is a typo.

3. As far as I understand, there are no error bars in the figures/tables. Please include standard errors.

---

> ### Author Response · Authors · 2024-11-24
> **Response to Reviewer s7iL**
>
> Thank you for the constructive and elaborate feedbacks. Please find our response below:
> > **1. Asymptotic marginal coverage [W1&Q1]**
>
> Thank you for suggesting the use of two calibration steps for CP. In case of misunderstanding, we evaluate the following two variants of PoT:
>
> Method 1(PoT-1). We equally split the total calibration set and then use the first subset to calculate $r^*$ (like PoT). The second calibration set is used to calculate the threshold $\hat{Q}$.
>
> Method 2(PoT-2). We equally split the calibration set into two subsets. The first subset is used to calculate $r^*$ (as in PoT). For the second subset, we exclude samples where the ground truth label's rank exceeds $r^*$. The remaining samples are used to calculate the threshold $\hat{Q}$.
>
> Both PoT-1 and PoT-2 construct their final prediction sets following the same approach as PoT. We evaluated these methods on the ImageNet dataset using Average Prediction Size (APS) across 10 independent trials.
>
> As shown in the tabel below, the base method, PoT, and PoT-1 achieve valid empirical coverage. Among these methods, **PoT achieves the smallest average set size across multiple models**. PoT-1 shows comparable efficiency to PoT and warrants further investigation in our future work.
>
>
> | Model       | Coverage (Base) | Coverage (+PoT) | Coverage (+PoT-1) | Coverage (+PoT-2) | Size (Base) | Size (+PoT) | Size (+PoT-1) | Size (+PoT-2) |
> |-------------|------------------|-----------------|--------------------|-------------------|-------------|-------------|---------------|---------------|
> | ResNeXt101  | 0.8993           | 0.8989          | 0.8988             | 0.8559            | 7.0472      | 6.5127      | **6.1848**    | 3.9441        |
> | ResNet152   | 0.9003           | 0.8995          | 0.8993             | 0.8562            | 6.3153      | **5.4801**  | 5.7849    | 3.9726        |
> | ResNet101   | 0.8992           | 0.8985          | 0.8990             | 0.8541            | 6.8317      | **6.2196**  | 6.4082    | 4.2553        |
> | ResNet50    | 0.8995           | 0.8987          | 0.8996             | 0.8617            | 9.0544      | 8.0018      | 8.4471    | 6.1013        |
> | DenseNet161 | 0.8988           | 0.8980          | 0.8987             | 0.8496            | 6.7598      | 5.8301      | **5.6994**    | 3.9808        |
> | Inception   | 0.9001           | 0.8998          | 0.8991             | 0.8435            | 35.9476     | 30.6982     | **29.6717**   | 12.9597       |
> | ShuffleNet  | 0.8994           | 0.8987          | 0.8982             | 0.8596            | 22.6550     | **21.0803**     | 21.7696   | 13.7674       |
> | **Average** | 0.8995       | 0.8989      | 0.8989         | 0.8544        | 13.9444 | **12.2604** | 12.9951   | 7.7116    |
>
> > **2. Asymptotic marginal coverage [W2]**
>
> When the oracle model is available, the ground-truth label would always be ranked first, resulting in $r^*$=1. However, in practice, the rank of the ground-truth label is unknown and unpredictable. Therefore, we can estimate $r^*$ more precisely with the calibeation set size n approaching infinity, where this optimal $r^*$ value need not be equal to K.
>
> Experimental results in Figures 3 and 5 demonstrate that **PoT consistently reduces the average size of prediction sets compared to standard CP**. This misleading trend can be explained by the calibration set size: when the calibration set is too small, the threshold (defined in Equation 1) may be imprecise, resulting in large reductions from PoT.
>
> > **3. Typos and sandard errors of results [Q2]**
>
> Thank you for the correction and suggestion. We have added the standard errors in our manuscript.

---

> ### Comment · Reviewer_s7iL · 2024-11-26
> **Follow-up**
>
> Thank you for implementing the PoT-1, which is the method I had in mind. The PoT-2 method is invalid due to the selection process. I strongly encourage the authors to prove that PoT-1 attains CP-like finite sample guarantees in this submission; this is in line with Weakness 2 raised by Reviewer GSMe.
>
>
> > When the oracle model is available, the ground-truth label would always be ranked first
>
> This statement is incorrect. For example, take a 3-class classification with P[Y=1|X=x] = 0.5, P[Y=2|X=x] = 0.4, P[Y=3|X=x] = 0.1. Then the "ground truth" label is a sample from this distribution. Here, w.p. 0.1 the rank can be 3 and the classifier is the oracle.
>
> In light of this, I kindly ask the authors to update their comment to my concern: I expect that for well-fitted classifiers, the value of r* would increase as the size of the calibration set increases.

---

> > ### Author Response · Authors · 2024-12-02
> > **Response to Reviewer s7iL**
> >
> > Thank you for your suggestion and correction. We will imporve the existing theoretical ananlysis in our future work.  Finally, we really appreciate the reviewer spending time to discuss with us. Thanks!

---

> > > ### Comment · Reviewer_s7iL · 2024-12-03
> > > **Reviewer reply**
> > >
> > > Thank you. I decided to keep score: a major revision is needed to (1) strengthen the theory, and (2) analyze the effect of the proposed method as the size of the calibration data increases.

---

### Official Review · Reviewer_AmRf · 2024-10-30

**Soundness:** 2
**Presentation:** 2
**Contribution:** 2
**Rating:** 6
**Confidence:** 4

**Summary:**

This paper aims to increase the efficiency of conformal prediction by reducing the set size. Unlike existing literature, this paper identifies the redundancy issue associated with CP in two separate scenarios: under coverage and over coverage. The inclusion of redundant labels leads CP to generate inefficient sets. To fix this problem, this paper develops a Post-Calibration Truncated Conformal Prediction (PoT-CP) procedure to eliminate class labels with high uncertainty from the prediction set for the sake of reducing the set size. Overall, this paper is well-written, and the conclusion is supported with solid evidences.

**Strengths:**

Extend the conformity score from one dimension to two dimensions, identify the redundancy in the class lalels included in the prediction set, develop a post-truncation conformal prediction algorithm to truncate prediction sets while maintaining coverage.

**Weaknesses:**

The motivation of this paper needs to be strengthend particularly the broad implications of reducing the set size of CP to the machine learning community and beyond needs to be clearly elaborated.

On the computational study side, I suggest reporting the average set size of PoT for under coverage and over coverage separately. As the set size of the under-coverage cases is always zero, if you could report the average set size of the two scenarios separately and as a whole, this will make it more direct to assess the impact of the proposed truncation operation.

It does not make sense when you claim "The key idea is to eliminate classes with high uncertainty in the prediction sets". What do you mean high uncertainty? Do you mean that 1 - class probability is the measure of uncertainty?

**Questions:**

1. Can you explain why the variation of calibration set size has a different effect on the average set size of different conformity scores as shown in Fig. 3?
2. It is also important to investigate the behavior of the conformal predictors when alpha takes other values. I am wondering what is the impact of increasing alpha value on the effect of the truncation operation in reducing the average set size.

---

> ### Author Response · Authors · 2024-11-24
> **Response to Reviewer AmRf (1/2)**
>
> We appreciate the reviewer's recognition and valuable suggestions. Please find our response below:
>
> > **1. Further clarification of motivation [W1]**
>
> We agree that the broader implications of reducing conformal prediction set sizes deserve more emphasis in our paper. We rephrase the second paragraph of Introduction as follows:
>
> > In addition to the marginal coverage, the length of the prediction set refers to the efficiency. Higher efficiency (i.e., smaller prediction sets) could speed up the decision-making process or assist users in better assessing the model's reliability. For instance, in medical diagnosis, smaller prediction sets enable doctors to make faster and more focused decisions while maintaining uncertainty awareness, and in autonomous driving, compact prediction sets allow systems to react more quickly while preserving safety guarantees. Thus, improved efficiency could enhance the practicality of CP in modern machine learning systems, bridging the gap between theoretical guarantees and real-world deployability.
>
>
> > **2. The average set size of under coverage and over coverage [W2]**
>
> Thanks for the suggestion. We analyze the prediction set sizes using four metrics: 'Size' (total average size), 'NoE' (average size excluding empty sets), 'CovS' (average size of sets containing ground-truth labels), and 'UCovS' (average size of sets excluding ground-truth labels). We conducted experiments on ImageNet using APS with significance level $\alpha$ = 0.1. As shown in the following Table, **PoT-CP effectively reduces both 'NoE' and 'CovS' while maintaining stable 'UCovS' values.** We add the empirical results to Appendix E of the revised manuscript.
>
> | Model       | Method | Coverage | Size     | NoE     | CovS     | UCovS    |
> |-------------|--------|----------|----------|----------|----------|----------|
> | ResNeXt101  | Base   | 0.8993   | 7.0472   | 7.2034   | 7.4099   | 3.8029   |
> |             | **+PoT**   |0.8989   | **6.5127**   | **6.6578**   | **6.8094**   | 3.8821   |
> | ResNet152   | Base   | 0.9003   | 6.3153   | 6.4943   | 6.5253   | 4.4153   |
> |             | **+PoT**   | 0.8995   | **5.4801**   | **5.6360**   | **5.6079**   | 4.3441   |
> | ResNet101   | Base   | 0.8992   | 6.8317   | 7.0239   | 7.1135   | 4.3180   |
> |             | **+PoT**   | 0.8985   | **6.2196**   | **6.3947**   | **6.4323**   | 4.3398   |
> | ResNet50    | Base   | 0.8995   | 9.0544   | 9.3257   | 9.4699   | 5.3348   |
> |             | **+PoT**   | 0.8987   | **8.0018**   | **8.2425**   | **8.2962**   | 5.4076   |
> | DenseNet161 | Base   | 0.8988   | 6.7598   | 6.9336   | 6.9994   | 4.6332   |
> |             | **+PoT**   | 0.8980   | **5.8301**   | **5.9802**   | **5.9754**   | 4.5567   |
> | Inception   | Base   | 0.9001   | 35.9476  | 36.0169  | 38.7808  | 10.4091  |
> |             | **+PoT**   | 0.8998   | **30.6982**  | **30.7580**  | **32.8949**  | 10.9887  |
> | ShuffleNet  | Base   | 0.8994   | 22.6550  | 23.1075  | 23.6533  | 13.7272  |
> |             | **+PoT**   | 0.8987   | **21.0803**  | **21.5016**  | **21.8941**  | 13.8393  |
>
>
> > **3. Clarification for uncertainty [W3]**
>
> Sorry for the misunderstanding sentence. In this work, we use non-conformity scores (e.g., 1-class probability or APS) to quantify prediction uncertainty. Specifically, for a given input $\boldsymbol{x}$, a higher non-conformity score $V(\boldsymbol{x},y_2) > V(\boldsymbol{x},y_1)$ indicates greater uncertainty in predicting class $y_2$ compared to $y_1$. Therefore, the truncated classes typically have higher non-conformity scores, reflecting greater prediction uncertainty. We have revised these explanations in the manuscript for clarity.
>
> > **4. Explanation of Figure 3 [Q1]**
>
> Since RAPS and SAPS are score functions with hyperparameters, these parameters may be less accurately tuned with small calibration sets. As the calibration set size increases, RAPS and SAPS can achieve smaller prediction set sizes due to better hyperparameter estimation. We have added experiments with smaller calibration sets (500 and 800 samples), which demonstrate that RAPS and SAPS produce larger prediction sets under limited calibration data.

---

> > ### Author Response · Authors · 2024-11-24
> > **Response to Reviewer AmRf (2/2)**
> >
> > > **5. Results across alpha values [Q2]**
> >
> > To investigate the effect of varying $\alpha$, we evaluate the average prediction set sizes using different score functions on ImageNet, with ResNeXt101 as the backbone model. As shown in the following table, **PoT-CP consistently achieves smaller average set sizes and lower SSCV values across different $\alpha$ values.** Additional results for other models under different significance levels are provided in Appendix E.
> >
> > | alpha | Method | Coverage | Size     | SSCV     | WSC      | CovGap   |
> > |-------|--------|----------|----------|----------|----------|----------|
> > | 0.01  | Base   | 0.9896   | 145.1734 | 0.0099   | 0.9872   | 1.5457   |
> > |       | +PoT   | 0.9895   | **119.9677** | **0.0094**   | 0.9872   | 1.5545   |
> > | 0.03  | Base   | 0.9699   | 42.2811  | 0.0260   | 0.9622   | 2.9695   |
> > |       | +PoT   | 0.9696   | **35.1892** | **0.0234**   | 0.9615   | 2.9883   |
> > | 0.05  | Base   | 0.9489   | 19.8068  | 0.0394   | 0.9417   | 4.1618   |
> > |       | +PoT   | 0.9486   | **17.5855** | **0.0345**   | 0.9413   | 4.1757   |
> > | 0.10  | Base   | 0.8993   | 7.0472   | 0.0678   | 0.8891   | 5.9361   |
> > |       | +PoT   | 0.8989   | **6.5127** | **0.0585**   | 0.8883   | 5.9533   |
> > | 0.20  | Base   | 0.8005   | 2.6037   | 0.1060   | 0.7860   | 7.7739   |
> > |       | +PoT   | 0.7984   | **2.0774** | **0.0684**   | 0.7843   | 7.8463   |
> > | 0.30  | Base   | 0.7017   | 1.5500   | 0.2732   | 0.6942   | 8.6969   |
> > |       | +PoT   | 0.6999   | **1.3786** | **0.1870**   | 0.6889   | 8.7349   |
> > | 0.40  | Base   | 0.6021   | 1.0913   | 0.3346   | 0.5955   | 8.8267   |
> > |       | +PoT   | 0.6011   | **1.0328** | **0.1218**   | 0.5933   | 8.8567   |
> > | 0.50  | Base   | 0.5007   | 0.8151   | 0.2206   | 0.4982   | 8.6467   |
> > |       | +PoT   | 0.5003   | **0.7989** | **0.1275**   | 0.4980   | 8.6523   |
> > | 0.60  | Base   | 0.3982   | 0.6091   | 0.1552   | 0.3954   | 8.1102   |
> > |       | +PoT   | 0.3982   | **0.6088** | 0.1552   | 0.3954   | 8.1102   |
> > | 0.70  | Base   | 0.2982   | 0.4410   | 0.2390   | 0.2966   | 7.2965   |
> > |       | +PoT   | 0.2982   | **0.4404** | **0.2161**   | 0.2966   | 7.2967   |
> > | 0.80  | Base   | 0.1989   | 0.2887   | 0.1161   | 0.1987   | 6.0805   |
> > |       | +PoT   | 0.1989   | **0.2887** | **0.1036**   | 0.1987   | 6.0805   |
> > | 0.90  | Base   | 0.0990   | 0.1425   | 0.0918   | 0.0989   | 4.4873   |
> > |       | +PoT   | 0.0990   | **0.1425** | **0.0843**   | 0.0989   | 4.4873   |

---

> > ### Comment · Reviewer_AmRf · 2024-11-27
> >
> > Regarding point 1, I am curious how to interpret the empty set when the proposed method is deployed in practical applications. When using CP, we have no idea which case/input the model will miss to cover. If you replace the original prediction set of CP with an empty set, how are we going to interpret the empty set to the decision maker?
> >
> > Regarding point 2, can you further elaborate on 'CovS' (average size of sets containing ground-truth labels), and 'UCovS' (average size of sets excluding ground-truth labels)? How come the UCovS differs from CovS so significantly?

---

> > > ### Author Response · Authors · 2024-12-02
> > > **Response to Reviewer AmRf**
> > >
> > > > Regarding point 1
> > >
> > > As the Equation 4, the construction of an empty set depends on standard CP rather than PoT-CP which truncates the redundant labels. Without PoT-CP, the standard CP still generates the empty set. Therefore, the decision-making system may seek human intervention when getting an empty set.
> > >
> > > > Regarding point 2
> > >
> > > The main cause of this significant difference is that the set including the true label has larger redundant labels than that of the prediction set excluding the true label. As shown in Figure 1(b), the redundancies of over-covered samples are significantly larger than those of under-covered samples.
> > >
> > > Finally, thank you for your suggestion and discussion again.  Your suggestions have significantly improved the manuscript's quality and clarity.

---

### Official Review · Reviewer_GSMe · 2024-11-01

**Soundness:** 2
**Presentation:** 3
**Contribution:** 2
**Rating:** 3
**Confidence:** 4

**Summary:**

This paper introduces a modification to the construction of conformal prediction sets in classification problems. While standard CP only requires finding the critical nonconformity threshold, the authors suggest an additional requirement: identifying a second threshold based on the rank of the nonconformity scores within their corresponding predicted distributions. This added criterion for constructing the set theoretically ensures that the prediction set will not expand, and it can empirically reduce the set size, as demonstrated in experiments. Furthermore, the authors theoretically show that, asymptotically, their approach will achieve the same marginal and conditional coverage as standard CP.

**Strengths:**

The paper is well-written and easy to follow. Exploring alternative methods for constructing prediction sets beyond the standard thresholding approach (on nonconformity scores) is indeed an intriguing direction. The fact that the idea can be generalized to any nonconformity function adds to its strength.

**Weaknesses:**

1. While it is interesting to explore different ways of constructing prediction sets beyond a simple thresholding criterion, CP offers a straightforward justification rooted in statistical hypothesis testing for rejecting a label from the set. It is unclear what statistical reasoning the proposed approach uses to reject a hypothesis (i.e., a label being in the prediction set) that traditional hypothesis testing could not reject.

2. In general, CP is a popular framework due to its ability to reason and provide finite-sample guarantees. Asymptotic results, while potentially useful when fundamental assumptions like exchangeability are violated, may not be particularly interesting.

3. I don't necessarily agree that having an empty set in CP is better than having a set with multiple labels but no coverage. If CP serves as an uncertainty representation tool, we naturally expect to see relative comparisons between the set sizes for different instances. Thus, a set with three labels is more uncertain than one with two labels, even if none of them cover the ground truth.

**Questions:**

1. SSCV, though used in other papers, is not a good measure of conditional coverage, and your experiments demonstrate this. In the best-case scenario, your approach should have the same conditional coverage properties as standard CP, so any improvement in such a metric clearly indicates its uselessness. Do authors agree?

2. To me, Figures 3.c and 3.d look weird. Can authors explain why they don't behave as APS and THR?

3. There are some typos in the proof of Lemma 1.

4. Theorem 3 is obvious and doesn’t really need to be presented as a theorem.

5. Have the authors considered comparing the efficiency by excluding the empty sets? Or have they compared the efficiency only for cases with coverage?

---

> ### Author Response · Authors · 2024-11-24
> **Response to Reviewer GSMe (1/2)**
>
> We appreciate the reviewer for the insightful and detailed comments. Please find our response below:
>
> > **1. Statistical Reasoning of PoT-CP [W1]**
>
> Our algorithm can be described as a two-stage hypothesis test. For each label $y$,
> 1. Stage 1:
> We set the null hypothesis as $H_0: Y_{n+1}=y$. The p-value is defined by $p(y)=\frac{\\{i=1, \ldots, n+1 \mid V_{i} \geq V_{n+1}\\}}{n+1}$. Then, if $p(y)>\alpha$, proceed to Stage2. if $p(y)\leq \alpha$, reject $H_0$.
> 2. Stage 2:
> We set null hypothesis as $H_{0,f}: V_f(X_{n+1},y)\leq r^*$, where $r^*$ can be seen as the $1-\beta$ quantile of $V_f$ scores with $\beta << \alpha$.  Then, if $V_f(X_{n+1},y)\leq r^*$, we include $y$ in final prediction set $C_T(X_{n+1})$.
>
> Than, the final prediction set: $C_T(X_{n+1})= \{y\in\mathcal{Y}| p(y)>\alpha , V_f(X_{n+1},y)\leq r^* \}$.
>
>
> > **2. Fundamental assumptions of CP [W2]**
>
> Thank you for highlighting this key point. Our asymptotic analysis provides valuable insights into the truncation behavior and efficiency quantification of PoT-CP. The empirical findings support our theoretical analysis: as shown in Figure 4 and Figure 5 in the revised manuscript, PoT-CP achieves comparable coverage guarantees with a small calibration set. Moreover, our asymptotic results require the i.i.d. assumption, which is standard for both machine learning and conformal prediction theory [1,2]. We acknowledge that extending our theoretical analysis to weaker conditions is an important direction for future work.
>
> > **3. Discussion about the empty set in our motivation [W3]**
>
> Thank you for pointing out the misleading sentence. We agree that relative comparison between the set sizes could present uncertainty in the prediction. However, in the manuscript, we assume that the ground truth label of a test example is known, thus we can judge whether the prediction set contains the ground truth label. If not, we can directly reject the predictive results from the model. Although the oracle setting is unavailable in practice, the trivial idea motivates the design of our approach. We have revised the relevant discussion in the paper to clarify this reasoning better.
>
> > **4. SSCV for conditional coverage [Q1]**
>
> We respectfully disagree that SSCV is not a useful metric. In our theoretical analysis, we prove that PoT-CP asymptotically achieves the same conditional coverage (both object-conditional and class-conditional) as standard CP. SSCV serves a different purpose by measuring coverage consistency across different set sizes. The improved SSCV results for PoT-CP complement, rather than contradict, our theoretical conclusions, suggesting that PoT-CP maintains theoretical guarantees while achieving better practical performance in terms of coverage stability.
>
> > **5. Explanation of Figure 3[Q2]**
>
> Thank you for pointing out the interesting phenomenon. Since RAPS and SAPS are score functions with hyperparameters, these parameters may be less accurately tuned with small calibration sets. As the calibration set size increases, RAPS and SAPS can achieve smaller prediction set sizes due to better hyperparameter estimation. We have added experiments with smaller calibration sets (500 and 800 samples), which demonstrate that RAPS and SAPS produce larger prediction sets under limited calibration data.
>
> > **6. some typos and reformulation of Theorem 3 [Q3&Q4]**
>
> Thank you for the correction and suggestion. We have revised them in the updated version.
>
> **References**
>
> [1] Anastasios Nikolas Angelopoulos, Stephen Bates, Michael Jordan, and Jitendra Malik. Uncertainty sets for image classifiers using conformal prediction. In International Conference on Learning Representations, 2020
>
> [2] Lei, Jing, et al. "Distribution-free predictive inference for regression." Journal of the American Statistical Association 113.523 (2018): 1094-1111.

---

> > ### Author Response · Authors · 2024-11-24
> > **Response to Reviewer GSMe (2/2)**
> >
> > > **7. The average size of sets by excluding empty and sets with coverage [Q5]**
> >
> > To further analyze the prediction set sizes, we define four metrics: 'Size' (total average size), 'NoE' (average size excluding empty sets), 'CovS' (average size of sets containing ground-truth labels), and 'UCovS' (average size of sets excluding ground-truth labels). We conducted experiments on ImageNet using APS with significance level $\alpha$ = 0.1. As shown in the following Table, **PoT-CP effectively reduces both 'NoE' and 'CovS' while maintaining stable 'UCovS' values.** We add the empirical results to Appendix E of the revised manuscript.
> >
> > | Model       | Method | Coverage | Size     | NoE     | CovS     | UCovS    |
> > |-------------|--------|----------|----------|----------|----------|----------|
> > | ResNeXt101  | Base   | 0.8993   | 7.0472   | 7.2034   | 7.4099   | 3.8029   |
> > |             | **+PoT**   |0.8989   | **6.5127**   | **6.6578**   | **6.8094**   | 3.8821   |
> > | ResNet152   | Base   | 0.9003   | 6.3153   | 6.4943   | 6.5253   | 4.4153   |
> > |             | **+PoT**   | 0.8995   | **5.4801**   | **5.6360**   | **5.6079**   | 4.3441   |
> > | ResNet101   | Base   | 0.8992   | 6.8317   | 7.0239   | 7.1135   | 4.3180   |
> > |             | **+PoT**   | 0.8985   | **6.2196**   | **6.3947**   | **6.4323**   | 4.3398   |
> > | ResNet50    | Base   | 0.8995   | 9.0544   | 9.3257   | 9.4699   | 5.3348   |
> > |             | **+PoT**   | 0.8987   | **8.0018**   | **8.2425**   | **8.2962**   | 5.4076   |
> > | DenseNet161 | Base   | 0.8988   | 6.7598   | 6.9336   | 6.9994   | 4.6332   |
> > |             | **+PoT**   | 0.8980   | **5.8301**   | **5.9802**   | **5.9754**   | 4.5567   |
> > | Inception   | Base   | 0.9001   | 35.9476  | 36.0169  | 38.7808  | 10.4091  |
> > |             | **+PoT**   | 0.8998   | **30.6982**  | **30.7580**  | **32.8949**  | 10.9887  |
> > | ShuffleNet  | Base   | 0.8994   | 22.6550  | 23.1075  | 23.6533  | 13.7272  |
> > |             | **+PoT**   | 0.8987   | **21.0803**  | **21.5016**  | **21.8941**  | 13.8393  |

---

> > ### Comment · Reviewer_GSMe · 2024-11-26
> >
> > Thank you very much for elaborating on my comments.
> >
> > regarding 1. and 2.  It remains unclear what the p-values for rejecting a label in the proposed approach are and whether they follow a uniform distribution, as is the case with standard CP. This might also explain why the authors were not able to provide any finite-sample guarantee for their approach.
> >
> > regarding 3. The reasoning here appears to be incorrect. The main assumption is that, for every x, there exists a ground truth conditional probability distribution P(Y∣X=x). This means the ground truth label is inherently a random variable. For instance, in a three-class classification setting with P(Y∣X=x)=[0.5,0.4,0.1], APS ideally outputs {1,2} as the prediction set to achieve 90% coverage. In 10% of cases, this set will not include the ground truth label (when the label is 3), but this does not imply the conformal set is incorrect.
> >
> > regarding 4. It is both theoretically and empirically impossible that your approach improves the **conditional coverage** property compared to standard CP. While it is certainly possible to define an arbitrary metric and outperform standard CP with respect to that metric, this does not indicate an improvement in conditional coverage. I would like to reiterate that the fact that the SSCV result for your approach is better than standard CP does not imply better conditional coverage. **This is rather a remark to the authors than a critique.**

---

> > > ### Author Response · Authors · 2024-12-02
> > > **Response to Reviewer GSMe**
> > >
> > > Thank you for the patient discussion.
> > >
> > > > Finite-sample guarantee
> > >
> > > In this paper, we proposed a new and effective CP method with an asymptotic guarantee. Moreover, asymptotic guarantee is also a common conclusion in the CP community [1,2,3]. We will further develop the currently proposed method to achieve finite validity in our future work.
> > >
> > > > Regarding 3
> > >
> > > Thanks for pointing out the concern. In our motivation (i.e., a toy analysis), when we know whether the prediction set contains the ground-truth label, the prediction set excluding the ground-truth label is equivalent to an empty set. Specifically, setting the prediction sets excluding the ground-truth label does not hurt the marginal coverage and conditional coverage.
> > >
> > > > Regarding 4
> > >
> > > As stated in Section "Introduction", our goal is to improve the efficiency with no harm for conditional coverage. Moreover, the results of experiments in the manuscript have verified the effectiveness of our method. In other words, our method mitigates the trade-off between efficiency and conditional coverage.
> > >
> > > Finally, we really appreciate the reviewer spending time to discuss with us. Thanks!
> > >
> > > [1] Chernozhukov, Victor, Kaspar Wüthrich, and Yinchu Zhu. "Distributional conformal prediction." Proceedings of the National Academy of Sciences 118.48 (2021).
> > > [2] Szabadváry, Johan Hallberg. "Beyond Conformal Predictors: Adaptive Conformal Inference with Confidence Predictors." arXiv preprint arXiv:2409.15548 (2024).
> > > [3] Xu, Chen, and Yao Xie. "Conformal prediction for time series." IEEE Transactions on Pattern Analysis and Machine Intelligence 45.10 (2023): 11575-11587.

---

### Official Review · Reviewer_P8xi · 2024-11-03

**Soundness:** 2
**Presentation:** 2
**Contribution:** 3
**Rating:** 5
**Confidence:** 4

**Summary:**

The authors empirically demonstrate that existing conformal prediction methodologies in classification problems lead to unnecessary inefficiency by including redundant labels, resulting in large prediction sets. To address this, they introduce a novel methodology called Post-Calibration Truncated Conformal Prediction (PoT-CP), which estimates a specific rank cutoff using calibration sets and applies truncation by assuming that beyond this rank, only redundant labels would appear in the prediction sets. The authors theoretically prove that their new prediction set converges to the original prediction set as the number of calibrations increases, thus establishing asymptotic marginal validity, and further discuss asymptotic conditional validity. Finally, this work justifies the applicability of their methodology through numerical experiments across various scenarios and datasets.

**Strengths:**

The most impressive strength of this work lies in its novelty. PoT-CP is motivated by a simple yet insightful observation, and the approach to address it is also straightforward. Rather than introducing a new non-conformity score or overhauling the prediction framework with complexity, the authors design an easily implementable algorithm by adding an ad-hoc procedure (i.e. truncation) applicable to various existing non-conformity scores.

Moreover, they conduct extensive numerical experiments across a range of models, datasets, and the choices of the non-conformity score. In particular, they thoroughly discuss and numerically demonstrate the advantages of PoT-CP when using non-conformity scores like APS, SAPS, and RAPS. This careful execution of exhaustive numerical experiments represents another notable strength of the work.

**Weaknesses:**

As the authors acknowledge in the discussion, the prediction sets derived from PoT-CP achieve asymptotic validity rather than finite-sample validity, meaning that the calibration set size significantly impacts PoT-CP’s coverage. However, the paper does not carefully address differences in coverage relative to data size. While they discuss changes in prediction set size with data size (lines 375–517), they omit discussion on empirical coverage.

Furthermore, despite the originality of their idea and the rigorous numerical experiments, the theoretical analysis appears somewhat sloppy and impedes readability, there are questionable statements in the theoretical analysis in terms of rigorousity. For instance, in Theorem 2 and Corollary 2, they discuss the almost sure convergence of (marginal) probability, even though probability itself is not defined on a measure space. Additionally, since PoT-CP does not achieve finite-sample validity, convergence rates should be discussed through careful use of such as Big-O notation; however, the paper lacks adequate explanation on this aspect.

**Questions:**

1. This question relates to the previously mentioned points. Did you observe how (empirical) coverage changes with variations in the calibration set size? If so, what were the results?

  2. The experiments in this work discuss only cases where the nominal level \alpha is fixed at 0.1. Did you find any systematic changes in the performance of PoT-CP when this value was varied?

---

> ### Author Response · Authors · 2024-11-24
> **Response to Reviewer P8xi (1/2)**
>
> Thanks for your recognition and valuable comments. Please find our response below:
>
> > **1. Empirical coverage across calibration set size [W1,Q1]**
>
> Thank you for highlighting this missing analysis. The empirical coverage rates across different calibration set sizes on the ImageNet dataset are presented in the following table. The results demonstrate that **PoT maintains comparable marginal coverage to the base method**. We have included the detailed results in Section 4.2 and Appendix E of the manuscript.
>
> | Score | Method | 500 | 800 | 1000 | 2000 | 4000 | 6000 | 8000 | 10000 | 13000 | 16000 | 18000 | 20000 |
> |-------|--------|-----|-----|------|------|------|------|------|-------|-------|-------|-------|-------|
> | THR   | Base   | 0.9024 | 0.8979 | 0.8969 | 0.8989 | 0.9017 | 0.8999 | 0.9003 | 0.9001 | 0.8997 | 0.9006 | 0.9003 | 0.9007 |
> |       | +PoT   | 0.8996 | 0.8951 | 0.8951 | 0.8973 | 0.9013 | 0.8997 | 0.9000 | 0.9000 | 0.8997 | 0.9005 | 0.9001 | 0.9007 |
> | APS   | Base   | 0.8971 | 0.8966 | 0.8953 | 0.8967 | 0.8993 | 0.8988 | 0.8992 | 0.8996 | 0.8992 | 0.9000 | 0.9000 | 0.9004 |
> |       | +PoT   | 0.8946 | 0.8952 | 0.8940 | 0.8960 | 0.8990 | 0.8985 | 0.8990 | 0.8995 | 0.8991 | 0.8999 | 0.8999 | 0.9004 |
> | RAPS  | Base   | 0.8907 | 0.8923 | 0.8922 | 0.8948 | 0.8983 | 0.8986 | 0.8995 | 0.8994 | 0.8994 | 0.8997 | 0.8999 | 0.9001 |
> |       | +PoT   | 0.8892 | 0.8912 | 0.8915 | 0.8941 | 0.8979 | 0.8983 | 0.8993 | 0.8992 | 0.8992 | 0.8994 | 0.8996 | 0.8997 |
> | SAPS  | Base   | 0.9010 | 0.8965 | 0.8957 | 0.8967 | 0.9007 | 0.9007 | 0.9005 | 0.8999 | 0.8996 | 0.9009 | 0.9001 | 0.9004 |
> |       | +PoT   | 0.8997 | 0.8959 | 0.8944 | 0.8964 | 0.9002 | 0.9004 | 0.9004 | 0.8996 | 0.8994 | 0.9006 | 0.9000 | 0.9002 |
>
> > **2. Discussion about convergence rates[W2]**
>
> We provide an analysis of the convergence rate as follows. To quantify the coverage rate, we use Hoeffding's inequality for bounded independent random variables. Let  $Z_{i}$, for  $i=   1,2, \ldots, m$ , be independent random variables where  $Z_{i}=1$  if  $E_{W_i}$  occurs given  $E_{V_i}$  on the $i$-th instance, and  $Z_{i}=0$ otherwise. The empirical estimate of the conditional probability is  $\widehat{\mathbb{P}}\left(E_{W} \mid E_{V}\right)=\frac{1}{m} \sum_{i=1}^{m} Z_{i}$ , which takes values in $[0,1]$. Thus, for any $\epsilon >0$, we can get that
> $$\mathbb{P}(|\mathbb{P}(E_{W}|E_{V}) -\widehat{\mathbb{P}}(E_{W}|E_{V}) | \geq \epsilon) \leq 2\exp{(-2m\epsilon^2)}.$$
> With $\widehat{\mathbb{P}}(E_{W}|E_{V}) =1$, we have
> $$\mathbb{P}(\mathbb{P}(E_{W}|E_{V}) \leq 1- \epsilon) \leq 2\exp{(-2m\epsilon^2)}.
> $$
> Let $2\exp{(-2m\epsilon^2)} = \delta$ and $m\approx (1-\alpha)n$. Then, $\epsilon = \sqrt{\frac{\ln{(2/\delta)}}{2n(1-\alpha)}}$. Assuming $\delta$ decrease polynomially with $n$, e.g., $\delta = n^{-k}$ where $k>0$. Thus, the convergence rate becomes:
> $$\epsilon = \mathcal{O} (\sqrt{\frac{\ln{(n)}}{n}}).$$

---

> > ### Author Response · Authors · 2024-11-24
> > **Response to Reviewer P8xi (2/2)**
> >
> > > **3. Results across alpha values[W3]**
> >
> > To investigate the effect of varying $\alpha$, we evaluate the average prediction set sizes using different score functions on ImageNet, with ResNeXt101 as the backbone model. As shown in the following table, **PoT-CP consistently achieves smaller average set sizes and lower SSCV values across different $\alpha$ values.** Additional results for other models under different significance levels are provided in Appendix E.
> >
> > | alpha | Method | Coverage | Size     | SSCV     | WSC      | CovGap   |
> > |-------|--------|----------|----------|----------|----------|----------|
> > | 0.01  | Base   | 0.9896   | 145.1734 | 0.0099   | 0.9872   | 1.5457   |
> > |       | +PoT   | 0.9895   | **119.9677** | **0.0094**   | 0.9872   | 1.5545   |
> > | 0.03  | Base   | 0.9699   | 42.2811  | 0.0260   | 0.9622   | 2.9695   |
> > |       | +PoT   | 0.9696   | **35.1892** | **0.0234**   | 0.9615   | 2.9883   |
> > | 0.05  | Base   | 0.9489   | 19.8068  | 0.0394   | 0.9417   | 4.1618   |
> > |       | +PoT   | 0.9486   | **17.5855** | **0.0345**   | 0.9413   | 4.1757   |
> > | 0.10  | Base   | 0.8993   | 7.0472   | 0.0678   | 0.8891   | 5.9361   |
> > |       | +PoT   | 0.8989   | **6.5127** | **0.0585**   | 0.8883   | 5.9533   |
> > | 0.20  | Base   | 0.8005   | 2.6037   | 0.1060   | 0.7860   | 7.7739   |
> > |       | +PoT   | 0.7984   | **2.0774** | **0.0684**   | 0.7843   | 7.8463   |
> > | 0.30  | Base   | 0.7017   | 1.5500   | 0.2732   | 0.6942   | 8.6969   |
> > |       | +PoT   | 0.6999   | **1.3786** | **0.1870**   | 0.6889   | 8.7349   |
> > | 0.40  | Base   | 0.6021   | 1.0913   | 0.3346   | 0.5955   | 8.8267   |
> > |       | +PoT   | 0.6011   | **1.0328** | **0.1218**   | 0.5933   | 8.8567   |
> > | 0.50  | Base   | 0.5007   | 0.8151   | 0.2206   | 0.4982   | 8.6467   |
> > |       | +PoT   | 0.5003   | **0.7989** | **0.1275**   | 0.4980   | 8.6523   |
> > | 0.60  | Base   | 0.3982   | 0.6091   | 0.1552   | 0.3954   | 8.1102   |
> > |       | +PoT   | 0.3982   | **0.6088** | 0.1552   | 0.3954   | 8.1102   |
> > | 0.70  | Base   | 0.2982   | 0.4410   | 0.2390   | 0.2966   | 7.2965   |
> > |       | +PoT   | 0.2982   | **0.4404** | **0.2161**   | 0.2966   | 7.2967   |
> > | 0.80  | Base   | 0.1989   | 0.2887   | 0.1161   | 0.1987   | 6.0805   |
> > |       | +PoT   | 0.1989   | **0.2887** | **0.1036**   | 0.1987   | 6.0805   |
> > | 0.90  | Base   | 0.0990   | 0.1425   | 0.0918   | 0.0989   | 4.4873   |
> > |       | +PoT   | 0.0990   | **0.1425** | **0.0843**   | 0.0989   | 4.4873   |

---

> > > ### Comment · Reviewer_P8xi · 2024-11-26
> > >
> > > I sincerely appreciate your efforts in thoroughly addressing my concerns regarding the additional experiments and the more careful theoretical analysis. I agree that your paper has become more convincing. On the other hand, I previously mentioned that a slight sloppiness in the theoretical analysis is unaddressed. While this is not a major that issue, particularly in Theorem 2 and Corollary 2, there seem to be minor problems with statements based on asymptotics. For example, in Theorem 2, there is a statement that the two events become equivalent as  $n \to \infty$  almost surely. While this is intuitively understandable, mathematically, the statement is a bit sloppy. Specifically, I believe that the following expression would be more rigorous:
> > >
> > > "Then, as  $n \to \infty$, we have  $Y_{n+1} \in \mathcal{C}(X_{n+1})$  is equivalent to  $Y_{n+1} \in \mathcal{C}_T(X{n+1})$  almost surely."
> > >
> > > This is because the left-hand side of Theorem 2 is defined as a marginal probability and is not defined with respect to a probability measure. (Alternatively, simply removing “a.s.” above the arrow $\rightarrow$ would suffice.)
> > >
> > > You might find this critique overly picky, but since this part represents the core theoretical results of your paper, I believe it should be represented as clearly as possible. Even though you have conducted sufficiently diverse additional experiments, it would be great if this aspect could be revised a bit further.

---

> > > > ### Author Response · Authors · 2024-12-02
> > > > **Response to Reviewer P8xi**
> > > >
> > > > Thanks for your correction. We will update the statement in Theorem 2 and Corollary 2 in the final version.  Your suggestions have significantly improved the manuscript's quality and clarity. We again appreciate your valuable feedback. Thanks!

---

> > > > > ### Comment · Reviewer_P8xi · 2024-12-02
> > > > >
> > > > > Thank you for making the correction. I would like to keep my score.

---

### Author Response · Authors · 2024-11-24
**General response**

We thank all the reviewers for their time, insightful suggestions, and valuable comments. We are glad that All reviewers appreciate that our work is **novel**, **intriguing** and theoretical (P8xi, GSMe). We are also encouraged that the reviewers find the motivation is **insightful** (P8xi) and the method is **generalized** with no hyperparameters (P8xi, GSMe, s7iL) and **plug-in** with no harm to existing methods (s7iL). Besides, reviewers recognize that the conclusion is **solid** (AmRf), supported by **extensive** and **thorough** experiments (P8xi, s7iL), and the writing is **well-written** and  **easy to follow** (GSMe, AmRf).

In the following responses, we have addressed the reviewers' comments and concerns point by point. The reviews allow us to strengthen our manuscript and the changes are summarized below:

- Added empirical coverage for different calibration set sizes in **Figure 4** and **Figure 5**. [P8xi]
- Added theoretical analysis on the convergence rate in **Section 3**. [P8xi]
- Added experiments on different $\alpha$ in **Appendix E**. [P8xi,AmRf]
- Clarified results for Figure 3 in **Section 4.2**. [GSMe,AmRf]
- Revised theoretical analysis in **Section 3**. [GSMe,s7iL]
- Added detailed analysis of prediction set size in **Appendix E**. [GSMe,AmRf]
- Revised motivation of CP in **Section 1**. [AmRf]
- Added error bar on the **all experimental results**. [s7iL]

For clarity, we highlight the revised part of the manuscript in blue color.

---

### Meta-Review · Area_Chair_UnQa · 2024-12-23

**Metareview:**

The paper propose a novel method that aims to improve the efficiency of conformal prediction by reducing redundant classes in prediction sets while preserving some desirable coverage. Redundant classes are those included in prediction sets despite having higher predictive uncertainty than the ground truth label. The paper introduces a redundancy metric to quantify this issue (reflects unnecessary inflation for the CP size) and numerically illustrate the claim existing conformal prediction methods may generate inefficient sets arising from the inclusion of redundant labels. The authors then propose truncating prediction sets to remove redundant classes based on their ranks and predictive uncertainty (mainly to align with the ground-truth). This is achieved using the Post-Calibration Truncated Conformal Prediction (PoT-CP) algorithm, which identifies and eliminates classes with excessive uncertainty while maintaining marginal coverage guarantees​.

The main issues with the rank of the ground truth label's non-conformity scores is in its sensitivity to calibration data and the choice of the conformity measure. If the calibration data poorly represents the true score distribution or the conformity measure is misaligned with the task, the rank may not accurately reflect the relevance of the ground truth label. Ambiguity in score ordering, particularly for close or tied scores, further complicates rank estimation (+ its computational cost). These issues risk misestimating the ground truth label’s rank, potentially causing failures in truncation processes that aim to optimize efficiency while preserving coverage guarantees. Perhaps that is why the authors insisted on asymptotic arguments.

The reviewers collectively raise concerns that some theoretical aspects of the paper may be overstated or imprecise, potentially leading to false claims (with specific counter example that authors failed to clearly address).

I overall recommend revisiting the paper and resubmit. I recommend a *reject*

**Additional Comments On Reviewer Discussion:**

- The guarantee of marginal coverage is only asymptotic, which reviewers GSMe and s7iL note undermines the practical appeal of conformal prediction (CP), which traditionally provides finite-sample guarantees. Without stronger theoretical justification, claiming robustness under small calibration sets may be misleading.

- Reviewer s7iL challenges the assumption that in an oracle setting, the ground-truth label is always ranked first, pointing out that in probabilistic scenarios, the rank of the true label can vary even with an optimal classifier. This misrepresentation could lead to erroneous conclusions about the theoretical behavior of the rank parameter  and its role in truncation.

- GSMe disputes the claim that the proposed method improves conditional coverage compared to standard CP. While the SSCV metric suggests practical advantages, this does not imply true conditional coverage enhancement. If the paper implies such improvement without rigorous evidence, it risks making a false claim.

- Reviewer AmRf questions the handling and interpretation of empty prediction sets in real-world scenarios. If the paper fails to address how decision-makers can act on such cases, it risks overgeneralizing the practical utility of the proposed truncation approach. The authors answer to this was quite fuzzy in my opinion.

- Reviewer s7iL notes that the second calibration method (PoT-2) presented in response is invalid due to its selection bias, which could misrepresent the method's practical validity.

---

### Decision · Program_Chairs · 2025-01-22

Reject